# Inflammation and Oxidative Stress in Diabetic Kidney Disease: The Targets for SGLT2 Inhibitors and GLP-1 Receptor Agonists

**DOI:** 10.3390/ijms221910822

**Published:** 2021-10-06

**Authors:** Agata Winiarska, Monika Knysak, Katarzyna Nabrdalik, Janusz Gumprecht, Tomasz Stompór

**Affiliations:** 1Department of Nephrology, Hypertension and Internal Medicine, University of Warmia and Mazury in Olsztyn, 10-516 Olsztyn, Poland; agatawiniarska@hotmail.com (A.W.); makb@poczta.fm (M.K.); 2Department of Internal Medicine, Diabetology and Nephrology, Faculty of Medical Sciences in Zabrze, Medical University of Silesia in Katowice, 41-800 Zabrze, Poland; knabrdalik@sum.edu.pl (K.N.); janusz.gumprecht@gmail.com (J.G.)

**Keywords:** type 2 diabetes, sodium-glucose cotransporter-2 inhibitors, glucagon-like peptide-1 receptor antagonists, diabetic kidney disease, oxidative stress, inflammation, autophagy, sirtuin 1, cytokines, macrophages

## Abstract

The incidence of type 2 diabetes (T2D) has been increasing worldwide, and diabetic kidney disease (DKD) remains one of the leading long-term complications of T2D. Several lines of evidence indicate that glucose-lowering agents prevent the onset and progression of DKD in its early stages but are of limited efficacy in later stages of DKD. However, sodium-glucose cotransporter-2 inhibitors (SGLT2i) and glucagon-like peptide-1 receptor (GLP-1R) agonists were shown to exert nephroprotective effects in patients with established DKD, i.e., those who had a reduced glomerular filtration rate. These effects cannot be solely attributed to the improved metabolic control of diabetes. In our review, we attempted to discuss the interactions of both groups of agents with inflammation and oxidative stress—the key pathways contributing to organ damage in the course of diabetes. SGLT2i and GLP-1R agonists attenuate inflammation and oxidative stress in experimental in vitro and in vivo models of DKD in several ways. In addition, we have described experiments showing the same protective mechanisms as found in DKD in non-diabetic kidney injury models as well as in some tissues and organs other than the kidney. The interaction between both drug groups, inflammation and oxidative stress appears to have a universal mechanism of organ protection in diabetes and other diseases.

## 1. Introduction

For the last two decades, the renal community has been mentally sliding into depression and nihilism. After publishing spectacular results of trials that discovered the reno-protective properties of drugs blocking the renin−angiotensin−aldosterone (RAA) axis, nephrologists have been painfully and repeatedly hit by several ‘negative’ studies. These included trials showing no benefit of a dual blockade of RAA, a correction of renal anemia or the use of erythropoiesis stimulating agents. Furthermore, there were trials demonstrating virtually no clinically meaningful outcome of any drug developed to treat mineral and bone disorders of chronic kidney disease (CKD-MBD), and the drugs saving the lives of other high-risk patients, such as statins and beta-blockers, seemed to provide no clear benefit in patients with moderately to severely advanced CKD (glomerular filtration rate (GFR) less than 45 mL/min/1.73 m^2^). The hope for patients came from diabetes research and practice, which was a light on the horizon. Indeed, a series of important clinical trials, initiated by the landmark studies EMPA-REG OUTCOME (with the sodium-glucose cotransporter-2 inhibitor (SGLT2i) empagliflozin) and LEADER (with the glucagon-like peptide-1 receptor (GLP-1RA) agonist liraglutide) demonstrated an unequivocal benefit in terms of cardio-vascular and renal event risk reduction and the prolonged survival of high-risk patients with different stages of diabetic kidney disease (DKD) [1,2]. Currently, there are studies in patients with diabetes across all GFR ranges that are being followed by landmark trials in which SGLT2i is also being used with great success in patients without diabetes [3,4,5]. SGLT2i were also shown to reduce the cardiovascular risk by means of decreased all-cause mortality, cardiovascular death, first hospitalization due to heart failure and decreased risk of combined endpoints composed of the above-mentioned events in patients with heart failure with reduced and preserved ejection fraction. These benefits were observed in patients with and without diabetes and were independent from age category (with a trend to achieve even better results in older patients), gender, BMI, and were evident across the entire spectrum of kidney function (to the values of GFR as low as 25 mL/min/1.7 m^2^) [3,4,5,6,7,8,9]. A recently published meta-analysis of four landmark cardio-vascular outcome trials on SGLT2i performed in patients with diabetes raised the possibility that this group of drugs may also protect from stroke and that this effect may be most pronounced for canagliflozin when used in patients with advanced DKD. The trend toward prevention of de novo atrial fibrillation or atrial flutter could also be noticed in analyzed trials [10]. Another meta-analysis that included seven outcome trials (with one also recruiting patients without diabetes) concluded that treatment with SGLT2i reduces the incidence of atrial fibrillation by 21% (Risk Ratio 0.79, 95% Confidence Interval 0.67; 0.93) [11]. The results demonstrating the impact of SGLT2i on cardio-vascular outcome in patients with heart failure and preserved ejection fraction regardless of diabetic status have been recently adopted by the recent European guidelines on heart failure in which empagliflozin and dapagliflozin gained the level of recommendation IA, regardless of diabetes [12]. In this review, we aimed to discuss the role of SGLT2i and GLP-1RA in moderating oxidative stress-mediated and inflammation-dependent renal injury in DKD. We also reviewed experiments in which the drugs from both groups were studied in kidney injury models other than diabetic kidney disease (for example ischemia/reperfusion injury, isoprenaline-induced renal failure or immune-mediated nephritis), where inflammatory and oxidative stress pathways were analyzed. In addition, we decided to discuss some experimental studies showing an interaction of SGLT2i and GLP-1RA with inflammatory and oxidative mechanisms of injury outside the kidney (with special attention paid to adipose tissue, heart and vessels, liver and circulating inflammatory cells). In our opinion, it is worth showing these extrarenal mechanisms of protection to demonstrate the universality of interference between the discussed drugs, inflammation and oxidative stress and the unequivocally beneficial effect on different tissues and organs. Such an approach also emphasizes the universal nature of tissue and organ damage in diabetes and other metabolic disorders. We intentionally decided not to discuss the experimental data on dipeptidyl dipeptidase 4 inhibitors. Paradoxically, these drugs are mirroring most if not all the benefits of GLP-1RA in experiments but have not been demonstrated as being effective drugs in outcome trials in terms of prolonging survival.

## 2. Inflammation and Oxidative Stress as Key Mechanisms of Kidney Injury in Diabetes

Traditionally, diabetic kidney disease (DKD) was considered a ‘metabolic’ kidney disease and was different than ‘inflammatory’ nephropathies, such as primary and secondary glomerular kidney diseases (many of them named ‘nephritis’ to emphasize their inflammatory nature). Now, it is very well recognized that the diabetic milieu triggers an inflammatory response and activates oxidative stress in tissues and organs affected by diabetes. The mechanisms leading to the excess stimulation of inflammation and oxidative stress were reviewed recently in a series of excellent papers [13,14,15,16,17]. In this review, we decided not to discuss them in a separate section but address them in more detail while describing their interactions with SGLT2i and GLP-1RA in particular studies.

## 3. SGLT2i as Antioxidative and Anti-Inflammatory Agents in DKD and Other Models of Kidney Injury

The notion that drugs designed to lower blood glucose may also have anti-inflammatory and anti-oxidative properties has been known for quite a long time. Metformin and pioglitazone are examples of drugs that decrease activation of macrophages and other cells involved in the activation of inflammation and the fueling of oxidative stress. These drugs promote the transition of proinflammatory M1 macrophages into the anti-inflammatory M2 phenotype but also limit the inflammatory response at the clinical level by means of lowering serum cytokine and acute phase protein levels [18]. Metformin is the mainstay of therapy in type 2 diabetes (T2DM), and it would be difficult to find the studies comparing the impact of metformin vs. other treatments on clinically meaningful patient outcomes. However, it seems that pioglitazone did not fully live up to promises based on its pleiotropic mechanisms of action; at best, it was found to be ‘neutral’ in terms of cardiovascular and renal outcomes.

The activation of the renin−angiotensin−aldosterone (RAA) axis is one of the key features of diabetes and an important pathogenetic factor that promotes tissue damage in this disease. Drugs interacting with RAA remain the mainstay of treatment in diabetes and DKD; in fact, most, if not all, clinical trials demonstrating the efficacy of SGLT2i and GLP-1RA were performed in patients who used the ‘newer’ drugs as an add-on therapy to the angiotensin-converting enzyme inhibitors (ACEi) or angiotensin II receptor blockers (ARB). The RAA story is evolving owing to the new generation of drugs interacting with mineralocorticosteroid-blocking agents, such as finerenone [19]. Since RAA may stimulate oxidative stress, inflammation and sympathetic activity, it plays an important role in the pathogenetic linkage of these three systems. The RAA axis is also linked with sodium-glucose cotransporters: angiotensin II (Ang II) upregulates SGLT1 and SGLT2 expression in several cell types, including endothelial and tubular epithelial cells [20,21,22,23]. Ang II was shown to upregulate expression of SGLT2 in immortalized human proximal tubular epithelial cells, and the angiotensin II receptor-blocking agent losartan was able to prevent this upregulation. Chronic Ang II infusion for 4 weeks was demonstrated to induce several histological lesions in murine kidneys, such as: glomerular sclerosis, mesangial expansion, tubule-interstitial fibrosis and intratubular cast formation. All these lesions were markedly attenuated when simultaneous treatment with canagliflozin was given (although the SGLT2 expression in renal tubules remained stable). Even more interestingly, in human kidney biopsies sampled from patients participating in the Nephrotic Syndrome Study Network (NEPTUNE) project (which actually did not involve patients with diabetes), strong positive correlations were found between mRNAs of SGLT2, angiotensinogen and renin within proximal tubular cells [20]. In non-diabetic rats, the chronic infusion of Ang II resulted in an elevation of blood pressure, proteinuria, loss of GFR and the development of histological indices of renal damage; moreover, at the cellular level, there was an increase in reactive oxygen species and the expression of SGLT2 mRNA and protein. In this model, upregulation of SGLT2 upon stimulation with Ang II resulted in a significantly increased uptake of glucose by tubular epithelial cells. All these lesions were prevented by adding losartan, but not empagliflozin; empagliflozin, however, potentiated the effect of losartan when both drugs were used together [21]. Satou et al. demonstrated an increase in angiotensinogen mRNA and protein expression in cultured tubular epithelial cells exposed to different concentrations of glucose in a dose-dependent manner. This expression was attenuated by adding canagliflozin (except for the highest concentration of glucose used in the experiment, i.e., 25 mM). The mRNA of SGLT2 increased in parallel with the increase in mRNA of angiotensinogen; this increase was also prevented by canagliflozin. Exposure of tubular epithelial cells to high glucose concentration also resulted in a significant increase in reactive oxygen species content within the cells. This increase was almost completely prevented by adding canagliflozin [22]. These experiments suggest a bidirectional interaction between the RAA system and SGLT2, with an increased expression of SGLT2 upon exposure to glucose increasing angiotensin synthesis, but on the other hand, Ang II increases the synthesis and expression of SGLT2 (one of the key physiological functions of Ang II in the kidney is to increase sodium reabsorption, and the main Ang II-dependent sodium reabsorption pathways are located in proximal tubules). All of the above data were also confirmed in other cells, critical for the development of vascular and renal lesions in the course of diabetes. Namely, exposure of endothelial cells harvested from porcine and rat arteries exposed to increasing concentrations of Ang II led to significantly enhanced and dose-dependent expression of SGLT1 and SGLT2. A simultaneous increase in reactive oxygen species content within the cells was also observed, which was abolished by adding the dual SGLT1/2 inhibitor sotagliflozin or SGLT2i empagliflozin. These data may suggest that SGLT2 activity contributes to the reactive oxygen species formation and oxidative stress upon stimulation with Ang II. Further experiments revealed that stimulation of type 1 receptors (AT1) for Ang II, NADPH (nicotinamide adenine dinucleotide phosphate) oxidase activity and SGLT1- and SGLT2-dependent mechanisms act in concert to induce endothelial dysfunction and a pro-coagulation state by means of a downregulated function of endothelial nitric oxide (NO) synthase, impaired NO synthesis and the increased synthesis of adhesion molecules, tissue factor, monocyte chemoattractant protein-1 (MCP-1) and beta-galactosidase activity (the latter serving as a marker of cellular senescence) [23].

Xu et al. demonstrated that the SGLT2i dapagliflozin prevents inflammation and ameliorates the loss of autophagic capacity of cultured human proximal tubular epithelial cells. These authors have shown that dapagliflozin acts on several levels. First, increased inflammation and decreased autophagy directly depend on an increased flux of glucose into tubular cells in a diabetic milieu. The treatment with dapagliflozin significantly decreased SGLT2 expression and glucose uptake by the proximal tubular cells. Reduced autophagic potential (with deterioration of cell function triggered by the ‘aging’ organelle that does not enter the process of ‘recycling’) is caused by glucose-dependent suppression of adenosine monophosphate-activated kinase (AMP-activated kinase, AMPK). Dapagliflozin promoted phosphorylation of AMPK, the first step in restoring normal autophagy. Simultaneously, the drug also decreased the activity of the mammalian target of rapamycin (mTOR), one of the key protein kinases that governs cell growth and proliferation, transcription and translation of several genes (including those encoding proinflammatory proteins and proteins governing cellular energy metabolism). mTOR together with AMPK are the elements of a nutrient sensing mechanism which becomes disrupted when nutrients are available in excess (as it happens in diabetes and obesity). The authors concluded that the restoration of autophagic flux in the proximal tubular epithelial cells upon treatment with dapagliflozin may at least in part depend on mTOR suppression. The cells exposed to high glucose were characterized by an upregulation of several proinflammatory cytokines, including interleukin 1β (IL1β), interleukin 6 (IL6) and tumor necrosis factor α (TNFα). Expressions of all mentioned cytokines within the cells as well as their concentrations in culture supernatant were significantly reduced upon treatment with dapagliflozin. Since transcription of the genes that encode proinflammatory cytokines is controlled by the master transcription factor nuclear factor κB (NF-κB), the authors tested the hypothesis that dapagliflozin may interact with this transcription factor. Indeed, they found that the p65 subunit of NF-κB is suppressed in proximal tubular cells exposed to high-glucose medium with dapagliflozin and that such a suppression is secondary to AMPK activation [24]. In summary, these authors were able to demonstrate the complex action of SGLT2i on several mechanisms involved in the control of autophagy and inflammation, and that the multilevel derangement of these mechanisms triggered by high glucose can be restored at several points by using dapagliflozin.

In an animal model of T2DM, dapagliflozin prevented the development of lesions typical for DKD, such as: glomerulomegaly, glomerular basement thickening, mesangial hypercellularity and tubular damage. In more detail, dapagliflozin in a dose-dependent manner prevented the expression of αSMA (alpha-smooth muscle actin) in glomerular mesangial and tubular epithelial cells (abundant expression of αSMA in kidney diseases is a marker of cell dedifferentiation and a change of the phenotype into the proinflammatory and profibrotic one). Treatment with dapagliflozin significantly decreased renal expression of TNFα. Diabetic animals were also characterized by significantly lowered renal expression of pentraxin 3 (PTX3), and treatment with dapagliflozin partially restored this expression [25]. The latter notion is very important, since PTX3 favorably attenuates the inflammatory activity of macrophages, promotes the M2 phenotype of these cells and downregulates NFκB, IL1β, TNFα and monocyte chemoattractant protein 1 (MCP1) [25,26].

Epithelial-to-mesenchymal transition (EMT) and endothelial-to-mesenchymal transition (EndMT) describe the process of de-differentiation of highly specialized cells into a mesenchymal lineage by means of polarity loss, rearrangement of the cytoskeleton and a loss of the cell—cell and cell—basement membrane interactions. The cells gain a migratory and invasive phenotype and may differentiate into other types of cells. This scenario contributes to several physiological processes, such as repair from injury and wound healing, and is also operational during embryogenesis; unfortunately, EMT and EndMT may also contribute to tissue scarring and fibrosis as well as promote cancer metastases [27]. The best-known example of EMT in renal disease is the de-differentiation of tubular epithelial cells upon stimulation with proinflammatory signals and excess proteinuria resulting in a change of phenotype towards myofibroblasts. These detach from the tubular basement membrane, migrate into the interstitium and—together with resident renal fibroblasts—start to synthetize excess extracellular matrix proteins and contribute to interstitial fibrosis of the kidney [28,29]. In the heart and vessels, EndMT contributes to cardiac remodeling resulting in both systolic and diastolic heart failure, atherosclerosis, arterial hypertension and pulmonary arterial hypertension, the progression of cardiac injury in valvular heart diseases and many other pathologies [30]. The ability of SGLT2i to interact with HIF1α was repeatedly mentioned throughout this review, and it is worth mentioning that hypoxia-induced expression of HIF1α promotes EMT by stimulation of transforming growth factor β (TGFβ)-dependent pathways and through activation of the PI3K/Akt signaling pathway. Understanding the role of HIF1α in EMT may allow for a better understanding of the role of SGLT2i in nephroprotection [31]. Given the well-established role of EMT in CKD and DKD progression and the spectacular nephroprotective and cardioprotective effects of SGLT2i and GLP1-R agonists, it may also be interesting to look whether these drugs interact with EMT and EndMT.

In their interesting experiment, Li et al. induced diabetes by STZ injection in CD-1 mice and found that diabetes led to advanced kidney fibrosis paralleled by an increased expression of such biomarkers of EMT as α smooth muscle actin (αSMA), smooth muscle 22α actin and vimentin. Empagliflozin protected the kidney from all these effects in opposition to insulin, which—normalizing glycemia in diabetic animals—did not prevent fibrosis and EMT. Treatment with empagliflozin, but not with insulin, significantly decreased the number of E-cadherin-positive/αSMA-positive cells and E-cadherin-positive/vimentin-positive cells (i.e., cells undergoing EMT) within renal tubules. The extent of EMT (expression of EMT markers) was paralleled by decreased Sirt3 expression and signs of aberrant glycolysis (as represented by an increased expression of hexokinase 1 and 2 and the M1 isoform of pyruvate kinase). Again, empagliflozin, but not insulin, brought all mentioned molecules to their normal expression levels, clearly indicating that restoration of Sirt3 signaling and amelioration of aberrant glycolysis are operational in the prevention of EMT and consequently in the protection from kidney scarring and fibrosis in diabetic kidney disease [32].

Empagliflozin was also evaluated as an agent that may influence the EMT process of renal proximal tubular epithelial cells in an in vitro model by Ndibalema et al. The authors cultured epithelial cells in a medium with normal or high glucose concentration, with or without empagliflozin (three different concentrations of the drug were used). The key purpose of the experiment was to clarify the possible role of SGLT2i in HIF1α regulation and to evaluate the possible role of HIF1α in protection of renal epithelial injury upon exposure to high glucose. It has been demonstrated that exposure to high glucose does not lead to an increase in HIF1α protein expression in tubular epithelial cells, whereas empagliflozin increases such expression in a dose-dependent manner. In addition, the increase in HIF1α was paralleled with an increase in GLUT-1 (with the exception of empagliflozin used in the highest concentration, i.e., 500 nM; the other concentrations were 50 and 100 nM). Empagliflozin downregulated expression of its own receptor, which was significantly upregulated upon exposure to high-glucose medium. Markers of fibrosis, namely fibronectin and collagen type IV, were significantly increased following incubation with a high-glucose medium; the increase in TGFβ, the key cytokine inducing EMT and interstitial fibrosis, was also observed. All these effects were prevented by adding empagliflozin, although a decrease in TGFβ was concentration-dependent, but reduction of fibronectin and collagen type IV content was similar for all three concentrations of the drug. The downstream mediator of TGFβ, Smad3, was also increased following exposure to high glucose and normalized by empagliflozin. The content of E-cadherin, a protein characterizing an epithelial phenotype of proximal tubular cells, was decreased following exposure to high glucose, and such a decrease was prevented by empagliflozin in concentrations of 100 and 500 nM but not 50 nM. However, high glucose increased the content of αSMA, but empagliflozin in concentrations of 100 and 500 nM prevented this increase. High-glucose medium stimulated proliferation of cultured tubular epithelial cells, whereas adding empagliflozin inhibited their proliferation in a concentration-dependent manner. The cited study did not directly answer the key question concerning the role of HIF1α as a mediator contributing to the protection of renal tubular epithelial cells from EMT and interstitial fibrosis (which has been acknowledged by the authors as a limitation of the study). Nevertheless, it provides several important, key pieces of data highlighting the role of SGLT2i in nephroprotection and points to HIF1α-dependent signaling as one of the pathways of such a protection [33].

A very interesting study that combined experimental data and human biopsy findings was published by Huang et al. The authors analyzed the content of STAT1, TGFβ1, fibronectin and collagen type IV mRNA and protein in human biopsy samples obtained from patients with diabetes and DKD, and control samples from non-diabetic patients who were subjected to surgery for renal cell carcinoma (healthy kidney from non-tumorous areas). The content of mRNA and protein of all mentioned proteins was highly and significantly increased in patients with DKD. Patients with DKD were characterized by an advanced tubulo-interstitial fibrosis, with accumulation of STAT1, fibronectin and collagen type IV in the regions of fibrosis. In the experimental part of the study, the authors induced diabetes in mice using STZ and evaluated the possible role of dapagliflozin in the prevention of renal lesions typical for DKD. They sacrificed mice not treated with dapagliflozin at different time points following diabetes induction (8, 12 and 16 weeks), and mice treated with dapagliflozin—after 16 weeks of diabetes. The content of STAT1, TGFβ1, fibronectin and collagen type IV mRNA and protein were gradually increasing in the kidneys of diabetic animals in consecutive time frames. Progressive tubulo-interstitial fibrosis was noted in weeks 8, 12 and 16, with fibronectin and collagen type IV accumulating in the regions of fibrosis. In addition, a time-dependent increase in tubular basement membrane thickness was observed in diabetic animals. Treatment with dapagliflozin completely abolished these changes, as assessed at week 16. Finally, the authors evaluated the expression of similar proteins and markers of EMT in cultured renal proximal tubule epithelial cells exposed to high-glucose medium in culture with or without the addition of dapagliflozin. High glucose increased the expression of STAT1, TGFβ1 and αSMA, whereas expression of E-cadherin was decreased in cultured cells (clearly indicating that cells entered EMT). Co-exposure to dapagliflozin prevented increase in profibrotic proteins and EMT markers and the decrease of E-cadherin. In conclusion, the authors were able to demonstrate using three different approaches that SGLT2i protects diabetic kidneys from progressive tubulo−interstitial fibrosis via an inhibition of the STAT1/TGFβ signaling pathway [34].

Das et al. examined several aspects of functional and phenotypic changes in cultured proximal tubular epithelial cells in a diabetic milieu and the possible effect of empagliflozin on these changes. First, they found that empagliflozin abolishes the superoxide generation in tubular cells in hyperglycemic conditions. The drug was shown to inhibit TRAF3IP2 (TRAF3-interacting protein 2), the proinflammatory protein induced by high glucose which activates IκB kinase (IKK)/NF-κB and promotes the expression of various inflammatory mediators. Indeed, the reduced activation of NFκB was also demonstrated in tubular cells exposed to high glucose medium and empagliflozin. p38-MAPK activation was also abolished by adding empagliflozin, and the effect of the drug was largely similar to that of a selective TRAF3IP2 gene knock-out, as well as to the use of two potent ROS scavengers: Tempol and Tiron. Exposure to empagliflozin was followed by a significantly decreased secretion of several proinflammatory cytokines, including IL1β, IL6, TNFα and MCP-1; again, these effects were identical to a TRAF3IP2 knock-out. Empagliflozin prevented high glucose-induced synthesis and release of matrix metalloproteinase 2 (MMP2), which is released by proximal tubular cells undergoing epithelial-to-mesenchymal transition, essential to digest tubular basement membrane and to migrate to the peritubular interstitial space. Key cellular markers of the mesenchymal phenotype, i.e., αSMA, fibronectin and vimentin, were upregulated by high glucose, whereas a diabetic milieu downregulated E- cadherin, the adhesion molecule anchoring tubular cells to the basement membrane. All of the above high glucose-stimulated events were prevented by adding empagliflozin. Finally, the experiment also demonstrated that empagliflozin prevents an advanced glycation end-product, which is mediated by the stimulation of TRAF3IP2 expression [35]. In summary, the discussed series of experiments allowed a description of several pathways in which the inhibition of SGLT2 may prevent tubulo-interstitial damage, mitigating inflammation, oxidative stress and EMT in proximal tubular epithelial cells.

Assembly and activation of inflammasomes is a critical step in the initiation of an inflammatory response to several stimuli. Diabetic milieu can induce inflammasome activity, which facilitates inflammation-dependent tissue and organ damage [36]. Birnbaum et al. hypothesized that diabetes in an animal model activates inflammasomes in the kidney leading to the development of DKD and that the SGLT2i dapagliflozin may favorably interact with such an activation. To verify this hypothesis, they assigned diabetes-prone BTBR ob/ob and C57BL/6J wild type mice (WT) into three treatment protocols where the animals were exposed to vehicle, to dapagliflozin or to dapagliflozin in combination with saxagliptin, which is a dipeptidyl peptidase-4 inhibitor (DPP4i). BTBR ob/ob mice developed clinical and laboratory features of diabetic nephropathy with glomerulomegaly (increased glomerular area) and mesangial expansion in the kidney histology. These unfavorable changes were largely attenuated by the use of dapagliflozin, and the protective effect was even more pronounced when dapagliflozin and saxagliptin were used together. Moreover, dapagliflozin alone, as well as in combination with DPP4i, significantly reduced elevated renal expression of mRNA of collagens type 1 and type 3 in BTBR ob/ob mice. Components of the inflammasome, i.e., mRNA of apoptosis-associated speck-like protein (ASC), nucleotide-binding oligomerization domain, leucine-rich repeat, pyrin domain (containing NLPR3 or NALP3), IL-6, IL-1β, caspase-1 and TNFα were significantly upregulated in BTBR ob/ob vs. WT animals, and both active treatments (i.e., dapagliflozin and dapagliflozin + saxagliptin) significantly decreased their expression. Using the two drugs in combination resulted in a significantly greater effect. Furthermore, the kidney levels of IL-1β, IL-6 and TNFα proteins were significantly higher in BTBR ob/ob, and, as in the case of the mRNA of inflammasome components, these levels were attenuated by dapagliflozin and to a greater extent when both drugs were used together (with no effect in WT mice). The kidneys of BTBR ob/ob animals were also characterized by higher protein levels of NLRP3 and caspase-3 and lower phosphorylated/total AMPK ratio as compared to WT mice. Dapagliflozin significantly decreased the level of NLRP3 and caspase-3 and increased the phosphorylated/total AMPK ratio. The NLRP3 and caspase-3 lowering effect was more prominent when saxagliptin was added, but it was not the case with the phosphorylated/total AMPK ratio increase. In addition, dapagliflozin and saxagliptin led to the reduction of increased concentrations of blood urea nitrogen (BUN), serum creatinine and cystatin C in BTBR ob/ob mice with no effect in WT mice. The described molecular changes influenced by dapagliflozin have a very important clinical meaning leading to the reduction of DKD progression in studied animals; adding saxagliptin augmented most of the described effects [37].

Another study on inflammasomes and SGLT2i in the kidney tissue was performed in C57BL/6J mice fed with a high-fat high-sugar (HFHS) diet and receiving empagliflozin at three doses: 1 mg/kg, 3 mg/kg and 10 mg/kg, or receiving no treatment. Animals on a regular diet served as control; in addition, the group of control animals receiving empagliflozin 10 mg/kg was also established. The HFHS diet induced kidney injury represented by severe vacuolar degeneration of proximal tubular cells on histological examination. Empagliflozin attenuated these lesions in a dose-dependent manner. With regards to inflammasomes, NOD-like receptor family pyrin (NLPR)3 protein expression and caspase-1 activation, as well as the content of IL1β (the end-product of NLRP3 inflammasome activation) were analyzed, and all were significantly increased in the kidneys of the HFHS diet-fed animals as compared to control mice. Caspase-1 activation and IL1β content were significantly decreased in empagliflozin-treated HFHS animals, and this effect was dose-dependent. Similar effects of empagliflozin on NLPR3 protein expression, caspase-1 activation and IL1β content were also observed in the liver (also dose-dependent, but statistically significant only for the highest empagliflozin dose); these changes were accompanied by a dose-dependent reduction in the degree of liver steatosis and triglyceride content in the liver. The HFHS diet also resulted in lipid droplet accumulation in cardiomyocytes and, as in the kidney and liver, this accumulation was dose-dependently mitigated by empagliflozin. Of note, in the case of cardiomyocytes, the HFHS diet did not induce activation of the NLPR3 inflammasome (i.e., no NLPR3 overexpression, caspase-1 activation nor IL1β content) [38].

The role of SGLT2 inhibition in the prevention of oxidative stress in tubular cells was explored by Hasan et al. The authors proposed a rat model of chronic renal injury induced by the administration of isoprenaline, a non-selective β-adrenergic receptor agonist that leads to chronic sympathetic nervous system stimulation, resulting in the activation of oxidative stress, inflammation and apoptosis which mimics the pathological features of oxidative kidney injury in humans. They studied the effect of this agent in animals who were treated with the SGLT2 inhibitor canagliflozin or only the vehicle. First, the authors documented that isoprenaline administration increases oxidative stress as measured by the increased content of malonyl dialdehyde (MDA), nitric oxide (NO) and advanced protein oxidation products (APOP) as well as increased myeloperoxidase (MPO) activity in renal tissue homogenates. It also increased the plasma levels of respective markers. In contrast, key members of the antioxidative defense system, namely the glutathione content, catalase and Cu/Zn superoxide dismutase activities in renal tissue homogenates and their levels/activities in plasma, were significantly decreased following exposure to isoprenaline. Isoprenaline also suppressed the activation (phosphorylation) of AMPK, Akt and eNOS. However, inducible nitric oxide synthase (iNOS) was upregulated. Virtually all of the undesired events that followed isoprenaline administration were prevented by concomitant use of canagliflozin: the drug augmented the antioxidant defense mechanisms; activated AMPK, Akt serine/threonine kinase 1 (AKT1) and eNOS; and decreased the formation of advanced glycation and oxidation products. Inflammatory cell infiltration in renal tissue and the percent area of fibrosis in renal histology were also markedly reduced following canagliflozin exposure in addition to isoprenaline administration. All of the mentioned protective mechanisms influenced by canagliflozin translated directly into full protection of renal function in studied animals exposed to isoprenaline [39]. Virtually all of the effects described above in the kidneys were reproduced by the same research group in isoprenaline-mediated cardiac injury. In addition, in heart, the authors documented that canagliflozin augmented erythroid 2-related factor 2/heme oxygenase 1 (Nrf2- and HO-1), which mediates antioxidant and anti-inflammatory signaling and thus translates into cardioprotective properties [40]. An interesting experiment proving the protective action of SGLTi against subclinical acute kidney injury (AKI) related to myocardial infarction was performed by Ahmed et al., who evaluated the influence of empagliflozin on renal oxidative stress in diabetic rats after the induction of myocardial infarction. Diabetes was induced using streptozotocin and myocardial infarction with left coronary artery ligation. The authors demonstrated that myocardial infarction leads to the development of subclinical AKI. It was defined as an elevation of kidney biomarkers, namely neutrophil gelatinase-associated lipocalin (NGAL) and kidney injury molecule-1 (KIM-1), but with normal serum creatinine levels and without a reduction of urine output. Moreover, subclinical AKI was associated with an elevation of renal tissue NADPH oxidase (NOX)-2, toll like receptor (TLR) 2 and 4, myeloid differentiation factor (MyD) 88, TNF-alfa, IL-1 beta and IL-18 mRNAs. Additionally, empagliflozin treatment was related to an increase in β- hydroxybutyric acid (βOHB) as an additional proof of its antioxidant action. All increases in markers of inflammation and tissue injury were mitigated by treatment with empagliflozin [41].

A very complex study on the impact of dapagliflozin on inflammation and oxidative stress in human proximal tubular epithelial cells was published by Zaibi et al. The authors cultured immortalized proximal tubular cells, and in the first part of the experiment, they exposed the cells to increasing concentrations of H_2_O_2_ (range: 0–2000 µM). In consequence, the extensive H_2_O_2_ concentration-dependent apoptosis of cells was observed. Next, after choosing a 200 µM H_2_O_2_ concentration for further experiments, it was demonstrated that adding dapagliflozin to H_2_O_2_-enriched medium protects cells in the culture from apoptosis and necrosis. Dapagliflozin was used in different concentrations (0.1, 1, 5, 10, 100 µM). Dapagliflozin in concentrations ranging between 0.1 and 10 protected cells against apoptosis, and in concentrations between 0.1 and 5, it protected against necrosis. The higher doses appeared to be toxic for the tubular epithelial cells. At a concentration of 200 µM, H_2_O_2_ increased reactive oxygen species (ROS) production by tubular epithelial cells with dapagliflozin having no impact on ROS production in basal conditions. It was virtually blunted when cells were exposed to H_2_O_2_. Of note, dapagliflozin in the lowest concentration, i.e., 0.1 µM, demonstrated a very beneficial impact on ROS generation, whereas concentrations of 1 and 10 µM remained ineffective. In more detail, mitochondria were shown to be the most important source of ROS following incubation with H_2_O_2_, and dapagliflozin decreased ROS generation in these organelles when applied at the lowest concentration of 0.1 µM. Since calcium ions are critically involved in the control of transmembrane transport, the authors analyzed the impact of dapagliflozin on the potential of a transient receptor cation channel, subfamily M, member 2 (TRPM2), which is a non-selective, H_2_O_2_-dependent calcium-permeable cation channel. Interestingly, dapagliflozin facilitated an influx of calcium ions into the cell via this channel when H_2_O_2_ was applied. At the same time, dapagliflozin did not influence the activity of other channels such as transient receptor potential channels 4 and 5 (TRPC4, TRPC5) and calcium release-activated calcium channel proteins 1 and 3 (ORAI1, ORAI3), which also facilitate ion transport into the cell [42]. The possible role of SGLT2 inhibitors in the regulation of calcium ion transport and intracellular calcium availability was also postulated for other cells, such as cardiomyocytes (see below) [43].

Epigenetic regulation by miRNA is an increasingly recognized mechanism that controls many biological processes. miRNA are small (20–25 nucleotides) non-coding RNA particles that modify the expression of target genes via promotion or inhibition of degradation of respective mRNAs based on their complementarity. Given their crucial impact on gene expression, they are now considered as new disease biomarkers and potential therapeutic targets or even therapeutic agents, and these new perspectives also apply to CKD and DKD [44,45,46]. For example, some miRNAs may interact with sirtuins, affect autophagy and apoptosis and directly influence the fate of the kidney cells and endothelial and vascular smooth muscle cells [47].

Some miRNAs (as for example miRNA-296) have the potential to influence the expression of SGLT2 and possibly to modify the function of SGLT2i in target tissues [48]. The same holds true for GLP1-R: for example, miRNA-665 downregulates GLP1-R in the hearts of rats with heart failure, and this mechanism may explain the miRNA-665-induced development of heart failure. The targeted inhibition of miRNA-665 restores expression of GLP1-R and AMPK signaling and prevents cardiomyocyte apoptosis as well as the development of heart injury [49]. miRNA-204 was demonstrated to decrease GLP1-R expression in pancreatic β cells, and deletion of miRNA-204 improved GLP1 responsiveness of these cells [50].

Empagliflozin was also shown to reduce high-glucose-induced NF-κB- and p38 MAPK-dependent induction of miRNA-21. Since miRNA-21 targets RECK (Reversion Inducing Cysteine-Rich Protein with Kazal Motif), the protein inhibiting the matrix metalloproteinase 2 enzyme that promotes EMT and cell migration, the interaction between empagliflozin and miRNA- 21 highlights another mechanism of renoprotection of SGLT2i [35].

## 4. SGLT2i as Antioxidative and Anti-Inflammatory Agents in Tissues and Organs Other than the Kidney

As was mentioned in the Introduction section of this review, according to our current understanding, similar mechanisms of injury are instrumental in different tissues and organs exposed to a hyperglycemic milieu. Hence, it is worth also looking closer at the mechanisms of SGLT2i in organs other than the kidney to better understand their protective role from the point of view of a nephrologist. It is especially important when considering the interplay between heart, liver, kidney and adipose tissue in such chronic diseases as T2D, CKD, non-alcoholic fatty liver disease (NAFLD), obesity, metabolic syndrome or heart failure [51,52]. Thus, looking at the action of SGLT2i, for example in the liver or heart, may bring important clues to a nephrologist as well; lessons learnt in other organs may also be important to better understand the pathophysiology of DKD and its treatment.

An abnormal, highly proatherogenic lipid profile is a hallmark of diabetes and becomes even worse in DKD. Lipid profile abnormalities are inseparably linked to liver disease. Furthermore, DKD and NAFLD develop and progress in parallel in metabolic disorders and share multiple pathogenic similarities [53]. Dapagliflozin was shown to prevent the development of an atherogenic lipid profile and decrease the amount of MDA, interleukins 1β and 18 as well as TNFα in liver homogenates obtained from rats fed with a high carbohydrate diet. Dapagliflozin not only prevented the increase in all mentioned biomarkers in the liver tissue but also ameliorated histological lesions typical for NAFLD [54]. Intriguingly, dapagliflozin was also able to protect the liver from alcohol-triggered damage. Treatment with this drug reduced alcohol-induced oxidative stress by means of a decreased level of MDA and sustained enzymatic activity of superoxide dismutase, and it decreased inflammation by downregulating the synthesis of interleukins 1β, 18 and TNFα, as well as an expression of the master transcription factor governing the inflammatory response, NF-κB. Dapagliflozin prevented a profound downregulation of Nrf2 and also PPARγ (peroxisome proliferator-activated receptor gamma) mRNA triggered by alcohol ingestion. Since Nrf2 and PPARγ interact with each other (Nrf2 stimulates transcription of PPARγ and vice versa) and potentialize their protective effects (for example, simultaneously decreasing expression of NF-κB), the notion that dapagliflozin may stimulate both proteins is of great clinical importance (especially given the fact that drugs specifically designed to boost Nrf2 and PPARγ, i.e., bardoxolone and glitazones, were not shown to be cardio- or renoprotective despite promising preliminary clinical reports) [55]. Exactly the same mechanisms of liver protection were also demonstrated for empagliflozin in an animal model of methotrexate-induced hepatotoxicity [56]. Promotion of autophagy, decreased apoptosis and inflammation (with mitigation of mTOR signaling and MCP-1 expression) were also shown in the livers of apolipoprotein E-deficient (ApoE^−/−^) mice fed with a high fat diet and treated with empagliflozin [57]. Taken together, it is tempting to speculate that SGLT2 inhibitors may represent universal protective pharmacological agents used to protect against different mechanisms of injury with the activation of inflammation and oxidative stress, regardless of the diabetic status and the target tissue.

In the first part of this review, we discussed experiments demonstrating the role of SGLT2i in the mitigation of increased activity of NLPR3 inflammasome in the kidneys and livers of diabetic animals using db/db diabetes-prone mice and animals with diabetes induced by using a high-fat high-sugar diet. Here, we would like to quote another study focused on NLPR3- inflammasome activity in the liver upon treatment with dapagliflozin in an experimental model of steatohepatitis in diabetes. Dapagliflozin was applied to diabetic, hyperlipidemia-prone apolipoprotein E-deficient (ApoE^−/−^) mice (to induce diabetes, the ApoE^−/−^ mice were additionally fed with a high-fat diet [HFD] and received streptozocin [STZ]). HFD/STZ in ApoE^−/−^ animals led to biochemical features of liver injury, an increase in liver mass and advanced liver steatosis. Liver lesions were more pronounced in HFD/STZ animals compared to mice receiving HFD alone (i.e., non-diabetic). Dapagliflozin prevented the increase in liver mass, attenuated biochemical abnormalities characterizing liver failure and improved liver histology in HFD/STZ ApoE^−/−^ (diabetic) mice. A profibrotic response was also noticed in ApoE^−/−^ mice receiving HFD and injected with STZ by means of increased staining for αSMA in the livers of diabetic animals. Treatment with dapagliflozin significantly attenuated such a response. The number of macrophages/monocytes infiltrating the livers of HFD/STZ ApoE^−/−^ animals was markedly reduced by treatment with dapagliflozin. Diabetic ApoE^−/−^ animals were also characterized by a higher liver content of MDA and dihydroethidium, markers of oxidative stress; furthermore, in the case of these two markers, treatment with SGLT2i led to their reduction. The authors evaluated the components of the NLRP3 inflammasome, i.e., NLPR3, caspase-1, IL-1β and IL-18. All were significantly higher in the liver tissue of non-diabetic ApoE^−/−^ mice vs. WT controls, but the highest values were found in diabetic (HFD/STZ) ApoE^−/−^ mice. Expression of NLPR3 and caspase-1 proteins as well as the release of mature interleukins 1β and 18 were markedly reduced in dapagliflozin-treated HFD/STZ ApoE^−/−^ animals. Treatment with dapagliflozin attenuated the activation of NLRP3 inflammasome especially in the animals with diabetes [58].

One of the key mechanisms of SGLT2i that ameliorates inflammation and oxidative stress is an ability to limit the unfavorable impact of an excess amount of nutrients on the cells and tissues (nutrient-sensing mechanisms). This is of special importance in the case of adipose tissue. Adipose tissue exposed to a nutrient excess increases synthesis of proinflammatory hormones and cytokines (including leptin, IL1-β and -6, TNFα). These mediators—when released from ‘inflamed’ adipocytes—have a direct damaging effect on several cell lines in the kidney, including: endothelial cells, podocytes, mesangial cells, and tubular epithelial cells. Obesity-related glomerulosclerosis (the clinical and pathological entity different from DKD) serves as an example of the disease originating from adipose tissue which is considered as developing secondary to inflammation and the action of adipocytokines on the kidneys. Excess nutritional supply decreases the capacity of cells to initiate and then to proceed with autophagy (the process of organelle recycling, physiologically activated by cellular starvation). Organelles that are not processed by autophagy when aged or damaged trigger cellular inflammation and eventually cell death (the damaged mitochondria and peroxisomes are the main sources of inflammatory and oxidative signals). Ketonemia with glucose loss occurring with the use of SGLT2i mimics the situation of starvation and thus stimulates autophagy. Restored organized recycling of organelles allows the avoidance of cell damage, excess inflammation and oxidative stress, resulting in improved cell viability and tissue protection. The key mediators of increased autophagic flux that become upregulated upon SGLT2 inhibition include: sirtuin 1 (SIRT1), fibroblast growth factor 21 (FGF-21) and the peroxisome proliferator-activated receptor γ co-activator 1α (PGC1α). Upregulation of SIRT1 leads to activation of hypoxia inducible factor 2α (HIF2α), the key element of hypoxia-sensing machinery within the cells. HIF2α controls transcription of several genes involved in the protection of the cells against hypoxia; the gene encoding erythropoietin is the best-known gene controlled by HIF. Owing to that, patients treated with SGLT2i experience clinically meaningful and sustained increase in hematocrit and hemoglobin concentration that cannot be explained by a transient and small reduction in extracellular volume following the treatment with these drugs. The rise in hemoglobin and hematocrit may per se be reno- and cardioprotective; even more importantly, this rise represents the activation of several cytoprotective pathways mediated by SIRT1, FGF21 and PGC1α that cannot be readily measured. HIF2α also has an anti-inflammatory potential and an ability to promote autophagic flux. As one can conclude, the action of SGLT2i mimics starvation and hypoxia at the cellular level, and these effects combined trigger several downstream adaptive mechanisms that result in cell and tissue protection. Currently, it is widely accepted that the mechanisms described above, linking the promotion of autophagic flux with decreased inflammation and oxidative stress, are essentially the same in the kidneys, heart and vessels [59,60,61,62,63,64]. The loss of calories and the weight loss effect that follow treatment with SGLT2i promote browning of adipose tissue and decrease its proinflammatory activity [59,64]. This issue has been studied by Xu et al., who treated male obese C57BL/6Jslc mice on a high-fat diet with empagliflozin and analyzed the distribution and composition of adipose tissue. The authors found that treatment with empagliflozin promoted brown adipose tissue formation even in animals with already established obesity. The use of empagliflozin allowed for significant attenuation of systemic inflammation as measured by decreased plasma levels of several proinflammatory cytokines (including IL6 and MCP-1) as well as inflammatory markers within the adipose tissue and the liver (by means of reduced expression of mRNA for such mediators of inflammation as p38-mitoge-activated protein kinase [p38-MAPK], NF-κB or extracellular signal-regulated kinase [ERK]). A shift in macrophage polarization from a proinflammatory M1 into an anti-inflammatory M2 phenotype was also observed in adipose tissue and the livers of mice treated with empagliflozin. The phenomenon of increased M1 to M2 polarization upon treatment with SGLT2i has also been identified as a factor that may prevent cardiac remodeling and fibrosis following myocardial infarction in animal models [65]. Lee and colleagues performed a well-designed in vitro study to highlight in more detail the mechanism of the anti-inflammatory effects of SGLT2i and the DPP4 inhibitor gemigliptin. These authors stimulated RAW 264.7 cultured murine macrophages with lipopolysaccharides (LPS) and co-incubated with empagliflozin and/or gemigliptin. Both drugs used in the experiment decreased the expression of mRNA for: TNFα, IL1β and IL6 in LPS-challenged macrophages (with no apparent difference between the two tested agents). The expression of CD80, a marker of the M1 macrophage phenotype, was also significantly decreased upon treatment with SGLT2i or DPP4i, and their concomitant use resulted in an additional decrease of this expression. The same observations were also made in relation to the mRNA expression of inducible nitric oxide synthase, cyclooxygenase 2 and prostaglandin E2 as well as several chemokines (CCL3, 4, 5, CXCL10), with a synergistic effect when the two drugs were used together. Finally, the experiment demonstrated that empagliflozin (and to a lesser extent gemigliptin) inhibited phosphorylation of NFκB, a signal transducer and activator of transcription (STAT) 1 and 3, Janus (JAK2) kinase and IKK. This very complex study allows a better understanding of the very complex mechanisms of the interaction between SGLT2i and macrophages which results in the attenuation of systemic inflammation through the downregulation of IKK/NF-κB, MKK7/JNK and JAK2/STAT1 pathways. The anti-inflammatory properties were greater when the two studied drugs were used in combination [66]. Similar results (i.e., a significantly decreased expression of mRNA for TNFα, IL1β and iNOS and a downregulation of NFκB together with an M1 to M2 phenotype shift) were also observed in macrophages obtained from the lungs of mice with LPS- induced acute lung injury treated with canagliflozin [67].

The burden of cardiovascular diseases in diabetes is of special interest to researchers investigating the mechanisms of action of SGLT2i in myocardium and endothelial cells. An intriguing mechanism of action has been shown for empagliflozin in the myocardium of obese mice induced by a high-fat diet. The drug was demonstrated to decrease the activity of mTOR; in addition, empagliflozin upregulated the Nrf2/HO-1 (nuclear factor erythroid 2-related factor/heme oxygenase 1) pathway, the key cellular system that controls and limits excess activation of oxidative stress and, by controlling NF-kB, downregulates inflammation [68]. It should be mentioned that several agents that upregulate Nrf2 were tested as potential drugs in experimental models of diabetic heart injury and DKD; the most promising, i.e., bardoxolone, did not enter into clinical practice despite certain renoprotective effects due to higher cardiovascular side effects as compared to placebo [69,70]. Although the story of bardoxolone in DKD and CKD has not yet been terminated, it seems that SGLT2i may be a much better choice in the treatment of chronic nephropathies due to a very good safety profile [71].

Kolijn et al. investigated the role of empagliflozin in the prevention of oxidative stress and inflammation, as well as the impact of SGLT2i on the function of isolated human cardiomyocytes derived from left ventricular (LV) biopsy samples performed on patients with heart failure with a preserved ejection fraction (HFpEF). Moreover, obese and lean Zucker diabetic fatty (ZDF) rats were used for animal experiments to investigate isolated rat hearts and endothelial cells. Upregulation of several inflammatory biomarkers (vascular cell adhesion molecule 1 [VCAM1], intercellular adhesion molecule 1 [ICAM1], IL-6 and TNF-α) was observed both in isolated human cardiomyocytes from HFpEF (healthy heart transplant donors served as a control in this part of the study) and in hearts of obese vs. lean animals. Empagliflozin added to cardiomyocytes in culture and as a treatment in obese rats significantly reduced expression of the above-mentioned markers in both models. All tested markers of oxidative stress, i.e., H_2_O_2_, 3-nitrotyrosine and lipid peroxide, were also elevated in the cytosol and mitochondria of cells in respective models, and empagliflozin added to cultured cells or as a treatment applied in vivo significantly reduced their concentrations. The authors discovered increased oxidation (i.e., activation) of PKGIα (protein kinase G type Iα, the cGMP-dependent protein kinase controlling the calcium flux within cardiomyocytes and thus the function of myofilaments) both in patients with HFpEF and in obese ZDF rats. Empagliflozin in respective models reduced oxidation of this enzyme, which has been interpreted as beneficial since its decreased activation has been linked to reduced cardiomyocyte stiffness and improved cardiac function [43].

Empagliflozin also exerted its protective function in LPS- stimulated cardiomyocytes: the drug significantly decreased the expression of TNFα mRNA and protein, the mRNA of inducible nitric oxide synthase and improved cardiomyocyte energy balance. Empagliflozin in the discussed in vitro model promoted a macrophage phenotype shift from M1 to M2. In parallel, the activation (phosphorylation) of AMPK was observed both in cardiomyocytes and macrophages. Finally, in the in vivo part of the experiment, LPS-injected mice were also evaluated using echocardiography and empagliflozin-treated animals were characterized as having a better ejection fraction [72].

In another very interesting experiment, Byrne et al. generated two models of heart failure in non-diabetic rodents: heart failure with reduced ejection fraction (HFrEF) generated by transverse aortic constriction surgery resulting in pressure overload, and HFpEF induced by a high-salt diet applied to Dahl salt-sensitive rats. Animals from both groups were randomly assigned to receive vehicle or empagliflozin. First, in the HFrEF model, HF further deteriorated in vehicle-only treated animals, whereas decline in ejection fraction was stopped in those receiving empagliflozin. Several other echocardiographic parameters worsened in mice with HFrEF, unless treated with SGLT2i. Interestingly, the observed beneficial effects were not associated with changes in body composition upon treatment with empagliflozin (including fat mass and lean mass). The detailed analysis of left ventricular heart sections revealed that empagliflozin (as compared to vehicle) significantly reduced transcript levels of NLPR3 inflammasome activity markers, such as Nlrp3, caspase-1, Nfkb and Tnfa; reduction of IL-18, IL-1b and Mac3 transcripts was also observed. Empagliflozin also prevented the development and progression of diastolic dysfunction in the model of HFpEF. The impact of a drug on the above-mentioned transcripts in heart sections was essentially the same in HFpEF as described previously for HFrEF [73].

A very interesting study has significantly contributed to the knowledge of an interaction between SGLT2i and the sirtuin pathway. It was published very recently by Onofrio et al. The authors collected atherosclerotic plaques during the endarterectomy procedure performed due to high-grade internal carotid artery stenosis in 227 diabetic and 296 non-diabetic subjects. Ninety-seven T2D patients were treated with SGLT2i prior to the procedure, with a mean treatment duration of 16 ± 4 months. The authors found that the expression of the SLC5A2 gene, encoding SGLT2, was significantly increased in T2D patients who never used SGLT2i as compared to non-diabetic subjects, and that patients treated previously with these agents were characterized by lower SLC5A2 gene expression as compared to the untreated. The same pattern of expression was also observed for SGLT2 and NFκB proteins: it was the lowest in non-diabetics, the highest in diabetic SGLT2i never-users and lower in SGLT2i users vs. never-users. However, the expression of sirtuin 6 (SIRT6) was highest in non-diabetic patients, lowest in diabetic SGLT2i never-users and significantly higher in SGLT2i users vs. never-users. As one would expect, plaques obtained from patients with diabetes were characterized with a higher content of macrophages and a higher expression of nitrotyrosine (a marker of oxidative stress), TNFα and MMP-9 (metalloproteinase 9, the key enzyme that destabilizes atherosclerotic plaques by means of extracellular matrix protein degradation) in comparison to non-diabetic patients. All mentioned lesions were attenuated in SGLT2i users. In addition to plaque analysis, the authors performed a series of elegant in vitro experiments in which they cultured human aorta endothelial cells (HAEC) with and without high-glucose (HG) medium and exposure to SGLT2i canagliflozin in different concentrations (exposure to high glucose was applied following incubation with canagliflozin). Using this experimental model, the authors found that HG medium significantly stimulates mitochondrial oxidative stress within the cells as well as intracellular and extracellular oxidative stress—all these effects were significantly attenuated when pre-treatment with canagliflozin was applied. Similar effects were also observed regarding the synthesis and release of the proinflammatory cytokines: IL6, IL18, TNFα and MCP1; all were stimulated by HG medium and pretreatment with canagliflozin attenuated their synthesis and release. Finally, the authors performed a similar experiment using siRNA that transiently silenced the SIRT6 gene. Silencing of the SIRT6 gene led to upregulated SGLT2 expression and treatment with SGLT2i partially counteracted this increase. Exposure to HG medium following previous silencing of the SIRT6 gene resulted in ‘massive’ (term used by authors) upregulation of SGLT2, and this effect was not prevented by canagliflozin. The release of proinflammatory cytokines, significantly increased in cells with a silenced SIRT6 gene, could not be prevented by exposure to SGLT2i. Similarly, oxidative stress induced by HG medium in the cells with their SIRT6 gene transiently silenced was not prevented by pretreatment with canagliflozin. The described fascinating experiments clearly point to SIRT6-dependent signaling as one of the key pathways exerting protection against oxidative stress and inflammation in atherogenesis. Even more importantly for the clinician, the authors performed a long-term outcome analysis of patients who underwent endarterectomy. In the multivariable Cox regression analysis adjusted for several clinical and laboratory confounders as well as differences in pharmacological treatment, the authors demonstrated that the SGLT2i users experienced significantly less major adverse cardiovascular events (MACE) in the two-year follow-up as compared to SGLT2i never-users; in addition, when all study patients were stratified according to the terciles of the SGLT2 plaque expression, the multivariable Cos regression analysis showed significantly shorter MACE-free survival for those with SGLT2 expression in the highest tercile [74].

Some data can also be found on the interaction between SGLT2i and miRNA in pathogenesis of diabetic cardiomyopathy. Zhang et al. found markedly increased expression of miRNA-30d in the hearts of diabetic rats with diabetic cardiomyopathy. Antagonizing miRNA-30d significantly improved the cardiac function of diabetic animals and prevented myocardial fibrosis. The same protecting results were also obtained by using SGLT2i. In the series of experiments, the authors were able to demonstrate that SGLT2i may improve cardiac function and prevent cardiac remodeling in the rat model of diabetic cardiomyopathy inhibiting miRNA-30d and thus promoting autophagy in cardiomyocytes [75].

Another study demonstrated that canagliflozin can inhibit the release of IL-6 from LPS-stimulated human coronary artery endothelial cells and that this release depends on the inhibition of hexokinase 2 (HK2) activity. HK2, a key enzyme engaged in a glycolytic pathway, is significantly upregulated in the course of diabetes; its increased activity is one of the effectors of aberrant glycolysis and leads (among other mechanisms) to mitochondrial dysfunction, oxidative stress and to the upregulation of MAPK (mitogen-activated protein kinase), an enzyme critically involved in the activation of genes that stimulate inflammation and apoptosis. Three SGLT2i widely used in clinical practice, i.e., canagliflozin, dapagliflozin and empagliflozin, were used in the experiment to evaluate their possible interaction with HK2. Interestingly, only canagliflozin was shown to inhibit HK2 activation, HK2-mediated ERK1/2 phosphorylation and IL-6 synthesis and to increase AMPK activation in LPS- stimulated human endothelial cells; dapagliflozin and empagliflozin did not induce these effects. This is an intriguing observation since the medical community perceives the three leading SGLT2i as equal in reno- and cardioprotective properties, whereas the cited study has demonstrated that certain modes of action on a molecular level may be the unique features of particular molecules (such as an ability to inhibit hexokinase in the case of canagliflozin) [76].

Aberrant glycolysis is the specific form of glycolysis which occurs under normal oxygen conditions (aerobic glycolysis) [77]. In DKD, fatty acid oxidation as a source of energy is defected, and that is why glycolysis serves as an alternative source of energy. Moreover, glycolysis supports fibrosis by fueling myofibroblasts with energy and EMT induction. It has been shown recently that restoration of SIRT 3 level by SGLT2 inhibitors leads to suppression of aberrant glycolysis [78]. SGLT2i therefore are expected to reduce hypoxic stress, and it may be hypothesized that on a long-term basis it would lead to prevention of progression of DKD (Figure 1).

Endothelial cells may also acquire a mesenchymal and then fibroblastic phenotype (EndMT) and thus contribute to scarring and fibrosis of the involved organs. Such a mechanism is operational in the chronic myocardial injury in diabetes. Tian et al. induced diabetes in Sprague−Dawley rats using STZ injection and found that diabetic animals develop systolic and diastolic dysfunction of the heart as well as hypertrophy of heart structures. Animals treated with dapagliflozin or metformin were characterized by significantly improved cardiac function and were protected from cardiac remodeling. Diabetes was associated with significantly increased myocardial expression of collagens type I and type III as well as vimentin, a well-known marker of EMT and EndMT. Upregulation of potent profibrotic cytokines, transforming growth factor β (TGFβ) and connective tissue growth factor (CTGF) was also noticed in myocardial tissue. Both drugs used in the experiment (dapagliflozin and metformin) almost completely alleviated all effects described above. The authors also cultured the neonatal rat fibroblasts and human umbilical vein endothelial cells (HUVEC) in the high-glucose medium and demonstrated all effects observed previously in vivo: upregulation of αSMA, TGFβ and downstream mediators of TGFβ signaling (Smad 1 and 2, Twist 1 and 2). As in the in vivo experiment, all above features of fibroblast activation and EndMT of HUVEC were ameliorated by dapagliflozin and metformin. Both drugs were shown to restore significantly diminished AMPK activity in high glucose medium; using siRNA or compound C to inactivate AMPK diminished all beneficial effects exerted by dapagliflozin [78].

Sepsis and cytokine storm represent the most extreme and clinically threatening types of systemic inflammation. Niu et al., in a model of LPS-induced sepsis in mice, found that the application of canagliflozin (used as a nasal spray or in an intragastric infusion) prevented the development of inflammatory lesions in a lung histology to an extent largely comparable to the effect of dexamethasone used in another group of animals. Both drugs reduced macrophage infiltration in the lungs to a similar extent. A broad spectrum of inflammatory cytokines, including IL-1α, Il-1β, Il-2, Il-4, IL-7–10, IL-12, IL-13, IL-17A, TNFα, MCP1, granulocyte-macrophage colony stimulating factor (GM-CSF) and interferon γ (IFNγ), were analyzed both in the lung tissue and in the serum. All listed cytokines except IL-7 were significantly elevated upon stimulation with LPS, and canagliflozin and dexamethasone led to a significant lowering of their tissue expression and serum levels to a similar extent (intragastric application of canagliflozin was more effective than the inhaled application). The findings suggest a multidirectional and complex anti-inflammatory action of canagliflozin, which appears to be as effective as dexamethasone. This allowed the authors to conclude that the inhibition of SGLT2 may be a useful strategy in the treatment of cytokine storm in severe COVID-19 infection [79]. With regards to the kidney, in a similar model of LPS-induced sepsis in mice, empagliflozin diminished the extent of sepsis-induced acute kidney injury (tubular injury), ameliorating an inflammatory response [80].

In this review, we focused on the experimental studies performed in vitro and in animal models exploring the impact of SGLT2 inhibition on inflammation and oxidative stress. However, it should be mentioned that several association studies were also performed in humans evaluating the link between SGLT2i and biomarkers of inflammation and oxidative stress. For example, in their excellent meta-analysis, Bray and colleagues identified and analyzed 23 trials on this issue and concluded that the treatment with SGLT2i consistently led to the reduction of the serum levels of CRP, IL-6 and TNF-α (assessed as biomarkers of inflammation), and 8-iso-prostaglandin F2α and 8-hydroxy-2′-deoxyguanosine [8-OHdG] (reflecting the burden of oxidative stress). Treatment with SGLT2i also resulted in an increased serum level of adiponectin, the cardioprotective cytokine derived from adipose tissue. The clinical trials summarized by the authors are in agreement with the findings from experimental studies discussed herein [81]. In our real-world observational trial, we were also able to find that treatment with various SGLT2i modifies the redox status and antioxidant enzyme activity in urine of patients with type 2 diabetes [82].

In Table 1, we summarized the selected mechanisms of interactions between SGLT2i, inflammation and oxidative stress in the kidney and other tissues and cells.

## 5. GLP1RA as Antioxidative and Anti-Inflammatory Agents in DKD and Other Models of Kidney Injury

Native GLP1, due to its very short half-life, cannot be used for treatment purposes. Hence, several modifications of this peptide have been attempted with great success in order to establish clinically useful drugs. Two types of GLP1RA are used in the treatment of diabetes and its complications: exendin-based derivatives and human GLP1 analogues [83]. Exenatide, a synthetic analogue of exendin-4, the natural GLP1RA, is the first peptide approved for the treatment of diabetes [84,85]. Exenatide was originally injected twice-daily; now, once-daily formulas are also available. This drug was followed by the next generation of drugs with a pharmacokinetic profile allowing for once-daily (liraglutide, lixisenatide) and once-weekly injections (semaglutide, albiglutide and dulaglutide) [86]. The first oral formula of the GLP1RA, known as semaglutide, has also been recently approved [87].

In one of the early trials in the field, rats with STZ-induced diabetes were treated with placebo or exendin-4. The kidney structure, function and several biomarkers of oxidative stress and inflammation in kidney tissue were assessed. In this complex study, it was demonstrated that the GLP1RA ameliorates elevated albumin excretion in diabetic rats and prevents a rise in serum creatinine. Glomerular size remained normal (glomerular hypertrophy typically observed in DKD was prevented) and no mesangial matrix expansion was noticed in exendin-4 treated diabetic animals. The drug significantly reduced macrophage infiltration in the kidney tissue. Exendin-4 effectively suppressed expression of multiple biomarkers of inflammation, as reflected by decreased mRNA for TGFβ, ICAM1 or CD14 in the renal cortex (all these markers were upregulated in diabetic rats not receiving GLP1RA). Moreover, staining for ED1-positive cells (macrophages) and ICAM1 protein as well as type IV collagen content were markedly reduced in animals treated with exendin-4. Concerning oxidative stress, 8-hydroxydeoxyguanosine content, mRNA of NADPH 4 oxidase (NOX4) and NOX4 protein expression were assessed, all being significantly upregulated in diabetic animals and significantly reduced with GLP1RA treatment. p65 NF-κB binding capacity, upregulated in diabetes, was significantly inhibited by the injection of exendin-4. In an in vitro part of the experiment, the authors cultured human monocytes in a normal and high glucose environment. They found that exposure to the high glucose environment increased the expression of genes encoding TNFα and IL1-β; adding exendin-4 significantly decreased expression of these genes. Exendin-4 also prevented TNFα-stimulated ICAM1 gene expression in cultured human glomerular endothelial cells [88].

A similar model (mice with STZ-induced diabetes) was also used in another study focusing on the possible interaction of GLP1RA with oxidative stress and inflammation in the diabetic kidney. Diabetes led to the development of several histological abnormalities in the renal tissue: decreased height and continuity of tubular brush border, vacuolization of proximal and distal tubular cells, necrosis of tubular and glomerular cells, areas of hemorrhage and mononuclear cell infiltration. All these lesions were significantly decreased when treatment with exendin-4 was applied. The histologic damage score was significantly lowered after treatment with GLP1RA. Exendin-4 almost completely restored the function of Na^+^-K^+^-ATPase, an enzyme essential for proper resorptive function of renal tubules, markedly decreased following the induction of diabetes. Several biomarkers of inflammation (TNFα, IL-1β), fibrosis (TGFβ, fibronectin) and chemokines attracting macrophages to the site of injury and promoting their infiltration (MCP-1, ICAM-1), as well as CD68 (a marker of macrophages), were significantly increased in diabetic kidneys. The expression of all the above-mentioned biomarkers was normalized when the treatment with exendin-4 was applied. It is worth adding that MCP-1 and ICAM-1 were produced by tubular epithelial cells (suggesting their phenotype shift into a myofibroblastic direction) and that CD68+ monocytes preferentially localized in the interstitial space close to these cells. Concerning the parameters of oxidative stress, reactive oxygen species production and MDA content were significantly increased in diabetic animals, and immunohistochemistry revealed an abundance of 8-hydroxy-2′-deoxyguanozine (8-OHdG), the marker of oxidative DNA damage. All these effects were prevented when the animals were treated with exendin-4 [89].

The beneficial influence of GLP1RA on oxidative stress was confirmed in a study by Liljedahl et al., who evaluated the kidney tissue proteome in healthy mice and in mice with STZ-induced diabetes, receiving vehicle or liraglutide. The authors observed that STZ injection led to a significantly decreased content of catalase and glutathione peroxidase-3, enzymes critically involved in tissue protection against oxidative stress. The abundance of both enzymes was restored when animals with STZ-induced diabetes were concomitantly treated with liraglutide [90].

Lixisenatide was able to ameliorate histological lesions induced by STZ injection in mice fed with a high-fat diet. This effect was parallel with the restoration of total antioxidant capacity of renal tissue, suppressed by diabetes induction, and the reduction of MDA upregulated by diabetes. Increased expression of renal iNOS and COX-2 was also normalized by the treatment with lixisenatide. TGFβ expression was also markedly attenuated when GLP1RA was used. Two different dosing regimens were used in the experiment, namely: 1 nmol/kg/day and 10 nmol/kg/day intraperitoneally. All beneficial effects of lixisenatide were more prominent when the lower dose of the drug was used. In addition, one group of animals was treated with glimepiride, and the effects of this sulphonyl-urea medication were in most aspects very similar to those achieved with low-dose lixisenatide [91].

Exendine-4 was shown to activate (promote phosphorylation) AMPK, which was inhibited in a mouse model of diabetes (C57BL/6J mice fed with a high-fat diet and then challenged with STZ). In the kidneys of diabetic mice, significant increases in the mRNA of TGFβ, type 1 collagen and fibronectin were observed, and these changes were prevented by exendin-4. Exendin-4 also protected diabetic animals from the development of histological lesions characterizing DKD. In addition, exendin-4 was able to inhibit high glucose-induced transition of cultured mesangial cells into the fibroblast-like phenotype (with concomitant inhibition of TGFβ1-signaling, the key pathway that activates kidney fibrosis) [92].

The model of STZ-induced diabetes was also used in another study that analyzed the impact of liraglutide on albuminuria and oxidative stress. Induction of diabetes significantly increased urinary 8-OHdG and MDA which increased in parallel with albuminuria. Superoxide generation was markedly elevated in diabetic animals and it was attenuated by liraglutide. mRNA of all analyzed components of the NADPH oxidase complex, i.e., NOX4, p22 phox, p49 phox and p91 phox, were markedly upregulated in glomerular homogenates obtained from the animals with STZ-induced diabetes. Respective proteins were also markedly increased in the Western blot analysis. Treatment with liraglutide normalized their expression to the levels comparable with control groups. TGFβ and fibronectin, markers of fibrosis, were upregulated in diabetic rats, and such an upregulation was prevented by GLP1R agonist. The inhibitory effects of liraglutide on superoxide generation and NADPH oxidase activity were also reproduced in vitro in the human mesangial cells cultured in high-glucose medium. The beneficial effects of liraglutide on oxidative stress in vitro were reversed by adding inhibitors of PKA and adenyl cyclase [93].

The impact of the GLP1R agonist exenatide, the Ang II receptor blocker olmesartan, or both drugs used in combination on renal function and biomarkers of oxidative stress was tested in insulin-resistant obese rats. Kidney NADPH oxidase 4 (Nox 4) expression, along with superoxide dismutase, catalase and glutathione peroxidase activities were measured in order to assess the oxidative stress in insulin-resistant rats receiving no treatment or treated according to one of the three protocols mentioned above. Nox4 protein expression was significantly higher, and the activities of the antioxidant enzymes were significantly lower in obese insulin-resistant rats as compared to lean counterparts. Each of the drugs used in the experiment significantly and to a similar degree decreased Nox4 expression in the kidney tissue of obese insulin-resistant rats (with no clear additive effect when the drugs were used together). Concerning enzyme activities, superoxide dismutase increased by 61% when exenatide was used (a non-significant increase), by 81% with olmesartan monotherapy and doubled when the two drugs were used together; catalase and glutathione peroxidase remained unchanged regardless of the treatment. To further assess the impact of three therapeutic approaches on oxidative stress, urinary 8isoprostane, renal nitrotyrosine (NT) and 4- hydroxy-2-nonenal (4-HNE) were measured. Olmesartan, exenatide and combined therapy reduced NT excretion to a similar extent (25, 22 and 27%, respectively) and combined treatment reduced urinary 4-HNE by 27%. These effects were parallel with a significant reduction of albuminuria in obese insulin-resistant drugs (by 45 and 55% in olmesartan and exenatide groups, respectively), and albuminuria was fully normalized when combined treatment was applied [94].

Advanced glycation end-products (AGE) are considered the key mediators initiating inflammation in the setting of diabetes. The proinflammatory action of AGE is mediated by their interaction with specific receptors (RAGE) on several cells involved in the inflammatory response, including macrophages. Mesangial cells form the scaffold for glomerular capillaries, produce and maintain the mesangial matrix, communicate with other glomerular cells by secreting soluble factors and may contribute to the glomerular capillary flow via their contractile properties. The essential roles they play in the integration of structure and function of the glomerulus may however change under pathologic conditions—mesangial cells may turn into key cellular effectors of glomerular damage. Namely, in several kidney diseases, they may proliferate, release excessive amounts of extracellular matrix (ECM) proteins (which eventually would result in glomerular sclerosis called DKD nodular sclerosis) and produce inflammatory mediators leading to the injury of neighboring glomerular cells. Several lines of evidence demonstrate that mesangial cells may—upon stimulation—obtain several features of macrophages [95]. Such a scenario takes place in diabetic glomerulus and is directly linked with RAGE expression on mesangial cells and their stimulation by AGE. Indeed, exposure of the cultured mesangial cells to AGE markedly increased their IL-6 and TNFα content, whereas concomitant exposure to exendin-4 or PPARγ agonist largely prevented an increase in both cytokines. Of note, these effects depended on the downregulation of RAGE on mesangial cells, induced by both types of treatment [96].

Huang et al. demonstrated that cultured human mesangial cells increase synthesis and release of type IV collagen, fibronectin and ECM proteins upon stimulation with high glucose. Liraglutide and exendin-4 were shown to blunt such an increase in a concentration-dependent manner. Liraglutide became ineffective when exendin 9-39, GLP1RA, was concomitantly applied. It has been demonstrated that store-operated calcium entry (SOCE), the mechanism in which the emptying of the endoplasmic reticulum calcium stores causes a calcium ion influx into the cytoplasm, regulates the synthesis and release of ECM proteins by mesangial cells. High glucose inhibited SOCE, and this inhibition promoted ECM synthesis, whereas liraglutide restored SOCE in the high glucose milieu. The role of SOCE in GLP1RA-mediated inhibition of ECM synthesis was confirmed by adding GSK-7975A, a selective inhibitor of SOCE, which blunted the beneficial effect exerted by liraglutide [97]. The same authors demonstrated that high glucose blocks the Wnt/β-catenin pathway in mesangial cells, thus leading to an increased synthesis of ECM proteins. Liraglutide was shown to increase Wnt/β-catenin signaling followed by a reduction of ECM protein synthesis. The results obtained in cultured mesangial cells were also confirmed in an in vivo model where reduced renal hypertrophy, mesangial expansion and glomerular fibrosis together with a reduced glomerular content of fibronectin, collagen type IV and α-SMA was demonstrated in diabetic rats treated with liraglutide [98].

As mentioned in the section discussing SGLT2i, excess apoptosis may represent one of the mechanisms promoting kidney injury in the setting of diabetes, whereas autophagy provides a rescue mechanism from progressive injury. SIRT1-controlled metabolic pathways seem to play an essential role in the regulation of the above-mentioned processes. Liang et al. analyzed the role of SIRT1 in liraglutide-dependent protection from kidney injury induced in a mouse model using a high-fat diet and in mouse mesangial cells cultured in high glucose medium. In an in vivo model, liraglutide allowed for decreased body weight and fat content as compared to vehicle-treated animals; serum fasting glucose and an intraperitoneal glucose toleration test were also normalized upon treatment with liraglutide. As expected, liraglutide prevented an increase in kidney weight (glomerulomegaly, i.e., hypertrophy of a single nephron, resulting in an increased size and weight of the kidneys, is a morphological hallmark of early diabetic kidney disease) and significantly decreased the urinary albumin-to-creatinine ratio. Lesions characterizing diabetic kidney disease that developed in animals that were fed a high-fat diet and could be seen using light microscopy and standard staining (namely, glomerular basement membrane thickening of glomerular capillaries, Bowman capsule thickening, tubular cell vacuolization and tubulointerstitial fibrosis) were prevented when liraglutide was injected. Homogenates of kidney tissue from animals on a high-fat diet assessed using Western blot analysis revealed significantly downregulated Bcl-2 and upregulated cleaved caspase-3 proteins (i.e., proteins that inhibit and execute apoptosis, respectively). The changes in the expression of apoptosis-controlling proteins were in parallel with significantly downregulated SIRT-1. Liraglutide injections prevented an increase in Bcl-2 and downregulated cleaved caspase-3; SIRT-1 also increased in liraglutide-treated animals. All these findings were also confirmed in an in vitro experiment, in which the mesangial cells were cultured in normal or high-glucose medium with or without liraglutide for 48 h. Cells exposed to high glucose medium expressed more cleaved caspase-3 and significantly less SIRT-1 as compared to the cells cultured in a normal glucose environment; these changes were mitigated when liraglutide was added to the culture medium [99].

A very complex study on the role of GLP1 in nephroprotection in the setting of diabetes was published by Fujita et al. in 2013. Two different mice strains were used (C57BL/6 Wild Type and C57BL/6 Akita mice; i.e., mice developing diabetes due to an insulin 2 gene mutation), and in both groups, there were animals with normal expression of GLP1R and animals with GLP1R deficiency (GLP1R^−/−^). The group of nephropathy-prone KK/Ta Akita mice was also studied. The animals from the latter group were injected with vehicle or liraglutide alone, or in combination with adenyl cyclase inhibitor SQ22536 or a selective PKA inhibitor H-89. Renal histology revealed that C57BL/6 Akita GLP1R^−/−^ mice developed marked mesangial expansion with a reduced podocyte number and increased fibronectin deposition along the glomerular capillary walls. Irregular thickening of the glomerular basement membranes was also observed; all lesions were much more pronounced when compared to both wild-type and C57BL/6 Akita GLP1R^+/+^ mice. Dihydroethidium (DHE) histochemistry and thiobarbituric acid-reactive substance (TBARS) assay were analyzed as markers of oxidative stress. DHE fluorescence and TBARS renal level were significantly increased in diabetic Akita vs. wild-type mice, and both indicators of oxidative stress were even more pronounced in C57BL/6 Akita GLP1R^−/−^ animals. NADPH oxidase Nox4 component is considered to be the main source of superoxide in diabetic kidney. Both NADPH activity and Nox4 expression were significantly higher in the kidneys of diabetic vs. wild-type animals. NO content was markedly reduced in diabetic GLP1R^+/+^ animals, and this reduction was even more pronounced in GLP1R^−/−^ mice as compared to wild-type mice (with no difference found between Akita GLP1R^+/+^ and GLP1R^+/+^ wild-type animals in NO and all mentioned markers of oxidative stress). All mentioned histopathology lesions described for the C57BL/6 Akita GLP1R^−/−^ animals were also present in nephropathy-prone KK/Ta Akita mice and were prevented by liraglutide injection (but not by the concomitant injection of liraglutide and SQ22536 or H-89). Indeed, liraglutide increased the renal activity of both adenyl cyclase and PKA as compared to vehicle-treated KK/Ta Akita mice, and these effects were abolished with concomitant use of SQ22536 and H-89. DHE fluorescence, TBARS renal level, NADPH activity and Nox4 expression were significantly higher, whereas NO content was markedly reduced in KK/Ta Akita mice. All these effects were corrected in liraglutide-treated animals (but not in those receiving liraglutide with SQ22536 or H-89) [100].

The interaction between GLP1RA and miRNA may represent another mechanism of protection from injury in DKD. Exendin-4, when added to high-glucose medium-cultured tubular epithelial cells, alleviated cell fibrosis as expressed by increased expression of fibronectin and collagen type 1 mRNA and protein. The drug was also able to decrease the number of extracellular vesicles (EV) released from cultured epithelial cells following exposure to high glucose (EV are lipid bound vesicles secreted by cells into the extracellular space containing different protein, lipid and genetic ‘cargo’ and having several biological purposes). EV released from epithelial cells exposed to high glucose were demonstrated to increase levels of fibronectin and collagen type 1 in recipient cells (i.e., cells incorporating released EV); if the EV-releasing (donor) cells were pre-treated with exendin-4, they failed to induce fibronectin and collagen type 1 in recipient cells. Further experiments have documented that EV from exendin-4-treated cells carried very little exendin-4, so the influence of the drug on recipient cells was unlikely. The authors hypothesized that the miRNA repertoire in EV, modified by exposure to high glucose and possibly normalized by co-exposure to exendin-4 may be the factor that respectively induces and inhibits the profibrotic changes in recipient epithelial cells. Indeed, it has been demonstrated that high glucose increases the expression of miRNA-192 in EV in a p53-dependent manner, and exendin-4 was shown to inhibit such an increase. An inhibition of miRNA-192 in donor cells blocked fibronectin and collagen 1 upregulation in recipient cells. In addition, miRNA-192 itself and EV derived from cells exposed to high-glucose medium were found to inhibit GLP1-R in recipient cells. Since a knock-out of GLP1-R also induced fibronectin and collagen 1 upregulation in recipient cells, it has been postulated that miRNA-192 promotes tubular epithelial cell fibrosis via inhibition of GLP1-R [101].

Ischemia−reperfusion injury (IRI) is a mechanism of renal damage not directly associated with diabetes; however, diabetic patients are much more vulnerable to this mechanism of kidney insult, for example in a course of acute kidney injury (organ reperfusion right after kidney transplantation and revascularization of tight renal artery stenosis may also represent examples of IRI). Evaluation of the possible role of GLP1RA prevention of the IR-induced kidney injury may also be of value for a better understanding of its mechanism in a diabetes setting since the mechanisms of injury and protection are likely to be universal. In a study by Chan et al., IRI was induced in rats by renal artery clamping for one hour. Rats were receiving DPP4i sitagliptin or exendin-4 (IRI rats receiving no medication as well as sham-operated rats served as comparator groups). In addition, some animals were given exendin-9-39, the antagonist of GLP1R. Animals were sacrificed 24 or 72 h following IRI induction. As might be expected, signs of severe tubular and glomerular injury were observed in IRI kidneys at both time points, and included cast formation, tubular dilatation, tubular necrosis and dilatation of Bowman’s capsule. These histological lesions were prevented in animals treated with sitagliptin and remained unchanged when sitagliptin with exendin-9-39 was applied. Histological lesions were quantified according to the injury score, which increased significantly in IRI only animals as compared to the sham-operated group. Lesions decreased in the groups treated with sitagliptin and were additionally attenuated when exendin-4 was used (both 24 and 72 h following IRI). IRI-exposed animals expressed a significantly higher protein content of oxidized protein, ROS, NOX1 and NOX2 (markers of oxidative stress), ICAM1, TNFα, MMP9 and NFκB (markers of inflammation) in the kidney tissue. An increase in these parameters was prevented by sitagliptin but not by sitagliptin + exendin-9-39. The mRNA expression of TNFα, MMP9 and IL-1β was significantly higher in IRI-only animals compared to sham controls and was decreased in sitagliptin and exendin-4 groups (with no difference between the two treatment groups). Concerning mRNA for PAI-1, the situation was similar, but using exendin-4 resulted in a more profound reduction of PAI-1 as compared to sitagliptin only. The same pattern was also seen for CD68+ macrophage infiltrates in renal parenchyma: the lowest number of cells were in sham-operated animals, and there was a marked increase in cells in IRI, a reduction of cells in IRI + sitagliptin as compared to IRI-only and a further reduction in animals receiving exendin-4. The mRNA expression of eNOS and IL-10 (a cytokine with anti-inflammatory properties) was higher in the exendin-4-treated group vs. sitagliptin-treated animals. GLP1R expression increased gradually from sham-operated through IRI only and IRI plus sitagliptin, reaching its highest value in IRI plus exendin-4 animals. Moreover, catalase and SOD expression reached higher values in exendin-4-treated vs. sitagliptin-treated animals. Indices of apoptosis were also corrected in IRI animals treated either with sitagliptin or exendin-4, with no substantial differences between the two drugs. Data obtained in this study indicate that both ways of intervening with GLP1R, i.e., a DPP4i blockade or a GLP1R stimulation, act similarly as agents preventing IRI; however, the action of GLP1RAs seems to be more ‘complete’ as compared to DPP4i [102].

Liraglutide was demonstrated to inhibit EMT in the experimental model of unilateral ureter obstruction (UUO) ligation of the ureter, resulting in hydronephrosis, which is a very potent signal that stimulates fast-progressing renal fibrosis resulting in irreversible kidney injury. Li et al. have demonstrated that UOO increased deposition of collagen within the parenchyma of an obstructed kidney and the increased expression of mRNA of fibronectin and collagen type I alpha 1, which are well-known markers of renal fibrosis, suggesting EMT of tubular epithelial cells. All these phenotypic changes are typically triggered by TGFβ. In the cited experiment, adding liraglutide ameliorated the expression and accumulation of all measured biomarkers of renal fibrosis; liraglutide directly inhibited the action of TGFβ, inhibiting expression of both TGFβ1 and TGFβ1 receptor. All these effects depended on GLP1 receptor stimulation since they were abolished when exendin 3, a GLP1-R antagonist, was applied [103].

Resident inflammatory cells and cells attracted to the kidney from the bloodstream, or even from the bone marrow, play an essential role in the development of several types of kidney injury, including diabetic kidney disease. Moschovaki Filipidou et al. tested the hypothesis that GLP1R inhibition or activation plays a role in regulating the inflammatory response and inflammatory cell infiltration in a murine model of nephrotoxic serum nephritis. The disease was induced in mice by an injection of rabbit Ig dissolved in incomplete Freud’s adjuvant, then followed by injection of heat-inactivated rabbit anti-mouse GBM anti-serum. This procedure resulted in the development of cellular crescents, fibrin deposit accumulation within Bowman’s space and endocapillary hypercellularity (a significant proliferation of cells within the glomerular capillary loops). Infiltrates composed of CD86+ macrophages, CD4+ T helper cells and polymorphonuclear granulocytes (PMN) were also observed. Proteinuria and a rise in serum creatinine were typical for rapidly progressing glomerulonephritis. Two models were used in the described experiment: mice without GLP1R expression (glp1r^−/−^) and wild-type animals. The clinical manifestations and kidney histology features of the disease were roughly similar in both groups of animals; however, the infiltrates with inflammatory cells were markedly more pronounced in glp1r^−/−^ animals. After looking at the mRNA expression of several markers of inflammation in lymph nodes and spleens, it was observed that glp1r^−/−^ mice are characterized by significantly increased expression of IL-10, IFγ, TNFα and Rorγt (RAR-related orphan receptor gamma, the marker of Th17 lymphocytes), as well as the transcription factors T-bet and Gata3 (orchestrating the Th1 and Th2 lymphocyte differentiation) as compared to wild-type mice. No difference in Tregs was noticed. The analysis of T-cell markers in kidney infiltrates revealed no difference in expression of markers characterizing Th1, Th2, Th17 or Treg between wild-type and glp1r^−/−^ mice. Lymphocytes isolated from the spleens of glp1r^−/−^ mice were characterized by significantly increased proliferation when challenged with LPS as compared to the wild-type animals. Finally, treatment with liraglutide in wild-type mice significantly decreased the development of histological lesions typical for nephritis; decreased the mRNA expression of collagen type I alpha 1 chain (the marker of fibrosis); reduced the infiltrates with PMN, macrophages, T helper cells and cytotoxic T lymphocytes; and reduced the gene expression of IFγ, TNFα and T-bet as compared to wild-type animals not receiving liraglutide. Gata3 and Rorγt gene expression did not differ between the treatment groups [104]. Although the cited study described the immune-mediated model of kidney injury, not DKD, we feel that the results are very important in understanding the role of GLP1R signaling in ameliorating inflammatory response in the kidney.

To conclude this section of the review, we would like to cite the series of outstanding experiments performed by Moellmann et al., which integrated several aspects of GLP1−GLP1R interaction in DKD. These authors focused not only on GLP1 but also on the cleavage products of this hormone appearing due to the enzymatic action of DPP4 with the formation of GLP-1 (9–37) and neutral endopeptidase (NEP, with formation of small fragments, including GLP-1 (28–37)). In contrast to GLP1, the cleavage products do not interact with GLP1R. The authors designed several experimental models. First, they overexpressed DPP4-resistant GLP1 or its cleavage products GLP1 (9–37) and GLP-1 (28–37) in db/db diabetic and control mice using adeno-associated viral vector. Second, they induced IRI by left kidney renal artery clamping for 30 min. Animals were treated with GLP-1, GLP1 (9–37), GLP-1 (28–37), exendin-4 and liraglutide; GLP1RA exendin-9 was also used in some experiments. Animals were sacrificed after 48 h following IRI induction. Finally, the authors analyzed human lymphocytes and mononuclear cells in culture while treated with the above-mentioned agents and also evaluated T-cell migration abilities following treatment with GLP-1 peptides. Treatment of diabetic mice with full-length GLP-1 significantly improved several aspects of glucose metabolism, but GLP-1 cleavage products (constructs) had no impact on the metabolic control of diabetes. Concerning the renal pathology, all GLP-1 peptides having no impact on glomerular lesions significantly improved or even normalized tubular lesions observed in untreated animals. This improvement was in parallel with reduced renal mRNA expression of the tubular injury markers *Kim1* (Kidney Injury Molecule 1), *Lcn2* (lipocalin 2) and the cytokines/chemokines *Tnfa* and *Ccl5* (chemokine (C-C) motif ligand 5, also known as RANTES (regulated on activation, normal T cell expressed and secreted)). All GLP-1 peptides reduced renal accumulation of macrophages, dendritic cells and CD3+ T cells. Hearts of animals were also examined and a reduction of T-cell accumulation was observed following treatment with all GLP-1 peptides. It is worth mentioning that overexpression of all peptides significantly prolonged survival in db/db mice. All GLP-1 peptides inhibited CCL-5-stimulated migration of T cells; this effect was not attenuated by exendin-9, the GLP1R antagonist. In the IRI model, pretreatment with any of the GLP1 peptides significantly reduced the renal infiltration of macrophages, CD4+ and CD8+ T-lymphocytes, and these reductions were not influenced by co-treatment with exendin-9. Of note, these changes were not accompanied by differential expression of tubular lipocalin 2 nor the differences with the degree of tubular injury (scored from 0 to 3). In the same model, liraglutide was able to reduce T-cell infiltration having no effect on macrophage accumulation, whereas exendin-4 did not impact T cell nor macrophage infiltration [105].

## 6. GLP1R Agonists as Antioxidative and Anti-Inflammatory Agents in Other Tissues and Organs

SIRT-1 is a target for GLP1RA in many tissues. For example, activation of SIRT1 and AMPK by exenatide in C57BL/6J mice fed with a high-fat diet led to the promotion of lipolysis, fatty acid degradation and mitochondrial biogenesis in the white adipose tissue of these animals. These changes were parallel with an increase in PPARα and PPARγ coactivator 1α (PGC1α). Exenatide was less effective in the *Sirt1^+/−^* mouse model which clearly points to SIRT1 being a critical factor mediating these beneficial effects on adipose tissue. Virtually all lesions observed in vivo were also reproduced in the cultured adipocytes exposed to exendin-4. Overall, based on these results, the authors concluded that changes induced by GLP1RA lead to browning of the white adipose tissue [106]. Similar results were also obtained by Góralska et al., who exposed cultured human adipocytes to exendin-4 and found upregulation of SIRT1 and SIRT3 gene expression, followed by mitochondrial bioenergetic stimulation [107].

Exendin-4 was shown to upregulate adiponectin mRNA expression and protein synthesis in cultured adipocytes. Adiponectin is a key anti-inflammatory cytokine (adipokine) released by the adipose tissue. This effect of exendin-4 was blocked by concomitant exposure of the cells to the selective blocker of protein kinase A (PKA), which clearly points to a PKA-dependent action of GLP1RA on adiponectin synthesis. As one could expect, an exendin-4-induced upregulation of adiponectin was also blocked by the concomitant use of GLP1RA. Of note, exposure of adipocytes to exendin-4 resulted in drastically decreased levels of IL-6 and MCP1 mRNA, which clearly documents a complex anti-inflammatory effect of GLP1RA in the adipose tissue [108]. Similar results (i.e., upregulation of adiponectin in cultured adipocytes exposed to exendin-4) were also obtained in another study. The effect of exendin-4 was blocked in this experiment when the cells were transfected with GLP1R-targeting siRNA. The study also demonstrated that exposure to exendin-4 increases translocation of the transcription factor Forkhead Box O1 (Foxo1) from the cytoplasm to the nucleus and that the silencing of Foxo1 by siRNA abolishes the exendin-4-induced upregulation of adiponectin synthesis. SIRT1, the mediator mentioned in this paper several times as an important target for SGLT2i and GLP1RA, promotes Foxo1-C/EBP (CCAAT-enhancer-binding protein) transcription factor complex formation and thus increases Foxo1-mediated adiponectin gene promoter activation. The authors targeted SIRT1 with siRNA and were able to suppress exendin-4-induced adiponectin synthesis. All these results were reproduced in an in vivo model of mice fed with a high-fat diet. This type of diet significantly decreased serum adiponectin as well as adiponectin mRNA and protein expression in adipose tissue, and the use of exendin-4 was able to prevent the decreased expression of adiponectin mRNA and protein (although a normal serum adiponectin concentration was not recovered). Changes in SIRT1 and Foxo1 expression were parallel with the changes observed for adiponectin: SIRT1 and Foxo1 were suppressed on a high-fat diet, and this effect was prevented by treatment with exendin-4 [109].

He et al. looked at the cytokine and chemokine profile in the sera of patients with type 2 diabetes and found that all tested biomarkers of systemic inflammation (i.e., IL-1β, IL-6, TNFα, RANTES and CXCL10) were significantly elevated as compared to healthy controls. In the in vitro experiment, they measured the concentration of the mentioned cytokines in supernatants of cultured peripheral blood mononuclear cells (PBMC) derived from diabetic patients and healthy controls and found significantly higher concentrations of IL-1β, IL-6 and TNFα in cultured cells obtained from patients with diabetes. Treatment with exendin-4 or specific inhibitors of ERK or p38 MAPK signaling pathways for 24 h significantly reduced cytokine levels in PBMC culture supernatants obtained from diabetic patients. In addition, exendin-4 significantly suppressed secretion of the chemokines CCL5 and CXCL10 from PBMC (these chemokines serve as chemoattractants recruiting inflammatory cells to the site of inflammation). In parallel with an increase in cytokines and chemokines, the superoxide content in supernatants was markedly increased, suggesting the role of oxidative stress in inflammatory activation of these cells. Flow cytometry analysis of peripheral CD4+ T lymphocytes and monocytes obtained from diabetic patients revealed markedly increased levels of phosphorylated (i.e., activated) kinases: phos-ERK and phos-p38 MAPK. Exendin-4 prevented such activation to an extent comparable with the specific inhibitors of both kinases [110].

Activation of PBMC was also analyzed in the cells harvested from the patients who received exenatide, the first approved GLP-1RA used for the treatment of diabetes, or placebo for twelve weeks. The cells were obtained from patients at time zero, and then after 3, 6 and 12 weeks following the treatment commencement. First, the analysis revealed a significant (by 22%) reduction of ROS generation within the cells from patients treated with GLP1RA vs. placebo—the difference was apparent at week 6 and remained statistically significant until the end of the treatment (week 12). In parallel, suppression of NFκB was also observed in cells from patients receiving exenatide (from the 6th week onward) as compared to placebo. This effect was followed by a significant reduction in the mRNA of TNFα and IL-1β, the major gene targets of NFκB. The mRNA of c-JUN N-terminal kinase (JNK1) and toll-like receptors (TLR) 2 and 4 were also suppressed in peripheral mononuclear cells from exenatide-treated patients. All these changes at the cellular level were mirrored by the trends in serum cytokine and inflammatory mediator levels: a significant reduction was noticed for MCP-1, MMP-9, serum amyloid A (SAA) and IL-6 (all differences vs. placebo were apparent 6 and/or 12 weeks after treatment initiation). ROS generation and biomarkers of inflammation were also analyzed at 2 and 4 h following a single injection of GLP1R agonist and demonstrated a significant decrease as compared to the injection of placebo [111].

In line with the cited experiments, Arakawa et al. investigated the impact of exendin-4 on inflammatory properties of circulating monocytes in two models: C57BL/6 mice and apolipoprotein A-deficient (ApoE^−/−^) mice. They found that adhesion of monocytes to the endothelial surface in C57BL/6 mice receiving GLP1RA was significantly reduced as compared to animals receiving saline injections. In ApoE^−/−^ mice, adhesion of monocytes to both the endothelial surface and to atherosclerotic lesions was reduced upon treatment. Reduction of monocyte adhesion was accompanied by significantly reduced expression of mRNA for several markers of inflammation, including: ICAM1, VCAM1, TNFα and MCP1 in these cells. In addition, the expression of NFκB p65 was significantly reduced in monocytes exposed to exendin-4. In the in vitro part of the experiment, the authors challenged cultured monocytes with LPS. Prior to such exposure, NFκB was not detectable in the nucleus and became abundant upon exposure to LPS. Concomitant treatment with exendin-4 prevented NFκB detectability in the nucleus, suggesting a blocking effect on the nuclear translocation of this transcription factor. Activation of the AMPK/PKA pathway was considered to be a key signaling pathway of the described anti-inflammatory properties of GLP1RA. It must be emphasized that an abundant presence of GLP1R was detected on monocytes and macrophages in the cited study [112].

The proportion between M1 and M2 macrophages (i.e., proinflammatory and anti-inflammatory regulatory and repair-stimulating phenotypes) changes in different stages of inflammation: in an acute phase (for example when the killing of invading microorganisms is a top priority), the M1 phenotype predominates, whereas the proportion of the M2 phenotype starts to increase when an acute response turns into tissue healing and repair. M1 and M2 macrophages differ in terms of cytokine repertoire, although the phenotype change from M1 to M2 or vice versa is not a ‘switch’ but a transition process, with intermediate stages between the two forms. Abnormal and/or prolonged polarization into the M1 phenotype may exaggerate the tissue injury and preclude recovery from inflammation. In the course of diabetes, several stimuli increase the M1 phenotype predominance, thus promoting inflammation, oxidative stress and target tissue injuries [113]. Induction of phenotype transition from M1 to M2 has been postulated for a long time as one of the promising methods in the treatment of inflammatory or malignant diseases. GLP1R agonists seem to fulfill such a promise. Shiraishi and colleagues demonstrated that GLP1 itself, as well as GLP1RA exenatide added to cultured human monocyte-derived macrophages, promoted the activation of a signal transducer and activator of transcription 3 (STAT3), and this activation was followed by a change in the cytokine repertoire in these cells (with increased expression of anti-inflammatory IL-10) as well as a shift towards surface antigens characterizing the M2 phenotype (i.e., CD163, CD204). Interestingly, the expression of IL-12 and TNFα remained unchanged. These phenotypic changes were completely abolished when GLP1R signaling was silenced. Co-culture of mouse macrophages with mouse-derived adipocytes resulted in a decreased synthesis and release of adiponectin, the antiatherogenic and anti-inflammatory adipokine from the latter cell population. If GLP1R agonist was added, the suppressing effect of macrophages on adiponectin synthesis by adipocytes would be reversed [114].

Studies demonstrating that GLP1RA attenuates the inflammatory response by direct impact on several cell populations engaged in inflammation are in line with early studies demonstrating that GLP1R knockout (glp1r^−/−^) in lymphocytes significantly increases their proliferation in response to mitogenic stimulation and leads to a decreased number of peripheral regulatory T cells (Treg), the key cell population responsible for moderation of the inflammatory response [115].

Liraglutide, another GLP1RA commonly used in the treatment of diabetes and obesity, was also evaluated as a modifier of inflammation and oxidative stress. Bruen et al. tested the anti-inflammatory properties of this drug in a model of apolipoprotein A- deficient (ApoE^−/−^) mice characterized by early development and fast progression of atherosclerosis. To accelerate the onset and progression of atherosclerosis, a high-fat and high-cholesterol (HFHC) diet was applied to one group of animals, whereas the other group received a low-fat zero-cholesterol (LF) diet. The mice were injected with escalated doses of liraglutide. In addition, the authors collected samples from human arteries of patients who underwent endarterectomy and then cultured the diseased plaque (DP) and relatively disease-free (RDF) portions of these samples. First, in a human model, the authors found that cultured DP samples released significantly more MCP1 as compared to RDF samples and that ex vivo treatment with liraglutide allowed for a significant reduction in MCP1 release. PBMC were also obtained from patients, and then cells were stimulated in vitro using LPS with or without liraglutide. Treatment with liraglutide significantly decreased the release of MCP1 and IL-1β from human PBMC and the expression of mRNA for TNFα within these cells. Furthermore, the trend toward increased secretion of IL-10, an anti-inflammatory cytokine, was also observed in cultured PBMC, reflecting the capacity of liraglutide to moderate the inflammatory response. In an animal part of the experiment, the authors demonstrated a significantly reduced total burden of atherosclerosis in whole aorta specimens from liraglutide-treated animals vs. those injected with only a vehicle. Significantly fewer macrophages infiltrating aortas were also found in high-fat high-cholesterol diet fed mice treated with liraglutide as compared to mice receiving the same diet and only vehicle. A non-significant trend was also noticed in the macrophage phenotype transition (from M1 to M2) in animal aortas. Cultured aortas from HFHC diet-fed animals released significantly more TNFα as compared to the vessels obtained from mice on an LF diet; liraglutide added to culture almost completely abolished the release of TNFα. An HFHC diet induced a significant shift between M1/M2 macrophage subpopulations as compared to an LF diet, with a predominance of the M1 cells in bone marrow and spleen. Liraglutide, having no impact on the number of mononuclear cells, reversed this shift towards M2 predominance both in the bone marrow and in spleen [116].

As we have mentioned already several times in this review, boosting autophagy flux (the recycling of ‘used’ organelles) remains one of the key mechanisms of action in the case of SGLT2i. GLP1 receptor stimulation was also shown to promote autophagy in many tissues. Liraglutide, the drug that significantly reduces the risk of cardiovascular events in high-risk patients with T2D and prolongs their survival, was shown to stimulate autophagy in cardiomyocytes derived from diabetic obese Zucker rats and in cultured neonatal rat cardiomyocytes. The cited experiment has demonstrated that treatment of the cells exposed to high glucose with liraglutide promoted autophagy and increased cell viability; the latter mechanism depended on enhanced activation (phosphorylation) of AMPK and inactivation of mTOR (liraglutide reversed the effects exerted on these two enzymes by high glucose). In the in vivo experiment, the improved cardiomyocyte viability and increased autophagic flux were parallel with significantly decreased myocardial fibrosis and on a functional level with improved cardiac function as assessed by echocardiography [89]. The same molecular mechanisms resulting in the promotion of autophagy were also demonstrated in cultured hepatocytes and in the in vivo model of high-fat diet-fed C57BL/6 mice in which liraglutide significantly reduced the degree of liver steatosis [90]. In both cited experiments, chloroquine (an inhibitor of autophagy) promoted heart and liver injury, and these effects were also prevented by liraglutide [117,118].

Another study performed in the cultured cardiomyoblasts pointed to the role of an exchange protein directly activated by cAMP (EPAC), mediating multiple actions of cAMP in cells. The authors demonstrated that exendin-4 inhibits methylglyoxal-induced oxidative stress in these cells. The study confirmed that exendin-4 increases cAMP content in cultured cells, decreases mitochondrial ROS generation and stabilizes mitochondrial membrane potential in cells subjected to methylglyoxal. GLP1RA prevented methylglyoxal-induced apoptosis of cardiomyoblasts and downregulated several genes involved in the execution of apoptosis. Using the specific inhibitor of PKA did not interfere with the antioxidative and anti-apoptotic action of exendin-4, whereas an inhibition of EPAC significantly attenuated the protective effects of this compound. The authors concluded that activation of EPAC, a downstream mediator of cAMP, is mandatory to exert the protective function of the GLP1R agonist by means of apoptosis mitigation. Akt activation through increased activity of PI3K was also inhibited by a selective EPAC inhibitor, whereas an EPAC activator led to Akt phosphorylation (as did the GLP1R agonist) [119].

GLT1-R agonists may exert a cardioprotective role in diabetic cardiomyopathy, and the protective pathways with regard to EndMT are similar to those described above for dapagliflozin. Yan et al. looked at the potential of the GLP1 analog exenatide to reverse EndMT in a mouse model of STZ-induced diabetes. Indeed, the authors found that diabetic mice developed heart failure. Extensive extracellular matrix protein deposition was observed in the hearts of diabetic animals, which was largely prevented by the GLP1 analog. The drug was also demonstrated to reduce the expression of von Willebrand factor and α smooth muscle cell actin (αSMA), the well-known markers of endothelial cell transition into fibroblastic phenotype, upregulated by the induction of diabetes. Expression of fibroblastic markers was also attenuated in an in vitro part of the experiment in which human artery epithelial cells (HAEC) were exposed to high-glucose medium. In addition, the GLP1 analog exenatide inhibited the migration potential of cells that entered EndMT. The protective action of GLP1 analog was demonstrated to depend on its ability to suppress poly(ADP-ribose) polymerase 1 (PARP1); the effects on the HAEC were identical following PARP-1 gene silencing. Inhibition of PARP-1 resulted in a significant decrease in ROS generation [120]. Tsai et al. used the same model of diabetes (C57BL/6 mice exposed to STZ); in the in vivo part of the experiment, they looked at the neointima formation in carotid arteries following wire injury in mice treated or not with liraglutide. GLP1-R agonist protected arteries from neointima formation and promoted re-endothelization following injury. The experiment revealed that injury-triggered neointima formation depends on EndMT stimulation, and significantly increased expression of αSMA in a vessel wall was observed. Such an increase was attenuated by the treatment with liraglutide. In the in vitro experiment, the authors demonstrated that an exposure of human umbilical vein endothelial cells (HUVEC) to high-glucose medium significantly downregulates E-cadherin expression (the protein that facilitates cell−cell interaction and epithelial/endothelial integrity) and upregulates expression of smooth muscle 22α actin, the marker of myofibroblastic phenotype. Liraglutide prevented these phenotypic changes by suppression of the TGFβ1 signaling. As in the case of several other mechanisms discussed for SGLT2i and GLP1-R agonists in this review, the protective mechanisms of liraglutide also depended on the AMPK signaling activation [121]. Intriguingly, the same mechanism, i.e., AMPK-dependent inhibition of EMT in renal tubular epithelial cells, was described some years ago for metformin, the drug that cannot be used in patients with advanced CKD but undoubtably has nephroprotective potential in earlier stages of the disease [122].

Exenatide treatment prevented the development of diabetic cardiomyopathy as assessed using echocardiography in T1D mice (the disease was induced by STZ) and T2D mice (the disease was induced with a high fat diet), although in T2D, the drug attenuated both systolic and diastolic left ventricular dysfunction, whereas in T1D, only diastolic dysfunction was improved. In both types of diabetes, the cardiac expression of catalase and manganese sodium dismutase (MnSOD) proteins were reduced—exenatide significantly increased expression of MnSOD in T1D and T2D and catalase in T2D. In turn, p53, a potent activator of apoptosis, was significantly upregulated in T1D animals and exenatide normalized its expression. As in the above-mentioned study, in an in vitro part of the cited experiment, the authors also focused on cultured cardiomyoblasts and reproduced most of the in vivo effects: reactive oxygen species production in these cells markedly increased in the high glucose milieu, and expression of MnSOD and catalase decreased. Exenatide added to the culture medium reversed both effects. High glucose induced apoptosis of the cells, and this effect was also prevented by exenatide (p53 increased upon exposure to glucose and decreased when exenatide was added). Adding catalase inhibitor to exenatide resulted in increased reactive oxygen production, an increase in p53 expression and the activation of apoptosis [123].

As in the case of SGLT2i, GLP1RA were also evaluated as anti-inflammatory agents and drugs that protect from oxidative stress in patients with diabetes. For example, Ceriello et al. studied a group of patients with T1D in the settings of experimentally induced, controlled hypoglycemia or hyperglycemia. The patients were divided into two matched groups: receiving GLP1 infusion or not. The following biomarkers were measured in patient plasma: 8-iso-prostaglandin F2α (8-iso-PGF2α, a marker of oxidative stress), sVCAM and IL-6 (as markers of inflammation); flow-mediated dilation (FMD) of brachial artery was assessed as a measure of endothelial function and FMD following sublingual nitroglycerine as parameter of endothelium-independent vasodilation. In both experiments (i.e., hypo- and hyperglycemia), serum 8-iso-PGF2a, sVCAM and IL-6 increased, whereas flow-mediated dilation decreased. In both circumstances, concomitant infusion of GLP1 prevented the observed changes. Endothelium-independent vasodilation remained unaffected by both experimental conditions [124]. This and other studies clearly show the direct translation of experimental in vitro and in vivo studies into human trials, and, even more importantly, into high quality clinical outcome trials [124,125,126,127,128]. An increasing body of evidence has recently allowed a meta-analysis of the randomized clinical trials in which the impact of GLP1RA vs. alternative treatment/placebo on the biomarkers of inflammation and oxidative stress were analyzed. Based on 40 included trials, it was concluded that GLP1RA significantly lowered serum C-reactive protein (CRP), TNFα and MDA, and significantly increased serum adiponectin. Trends toward serum leptin, IL-6, and plasminogen activator inhibitor 1 level lowering were also observed but did not reach statistical significance. GLP1RA led to lowered serum MDA but did not influence other analyzed markers of oxidative stress, i.e., serum 8-iso-PGF2α and urinary 8-hydroxy-2′-deoxyguanosine. This meta-analysis is in our opinion very important since it confirms most of the findings from experimental studies in a significant number of high-quality clinical trials [129]. Unequivocally positive results were also found when only one acute-phase protein, CRP, was included in the meta-analysis: GLP1RA significantly decreased serum CRP in patients with diabetes and different co-morbidities, such as: coronary artery disease, obesity, hypertension, non-alcoholic fatty liver disease and psoriasis [130].

Such a broad spectrum of multidirectional anti-inflammatory and anti-oxidative incretin-based therapies, especially GLP1R agonists, opens new avenues for their use in other diseases, including autoimmune disease (as reviewed for example by Ranbdaksh et al. and McLean et al. with respect to autoimmune thyroid diseases, inflammatory bowel diseases, psoriasis, rheumatoid arthritis, late autoimmune diabetes in adults [LADA], asthma, multiple sclerosis and psoriasis) [131,132]. Looking at the contemporary literature, it appears that basic research focused on the influence of incretin-based therapies on inflammation has developed most progressively in the area of neurodegenerative diseases, and these efforts of scientists are quickly applied in practice and translated into protocols of clinical trials. The results of these trials are impatiently expected [133]. In this review, we intentionally decided not to discuss this very large and fast-growing area of research on GLP1 and GLP1RA, but it seems that neuroprotective mechanisms of the incretin-based therapies are very similar to those implicated in the protection of other tissues and organs.

Table 2 summarizes the selected mechanisms of interactions between GLP1RA, inflammation and oxidative stress in the kidney and other tissues and cells.

In Figure 2, we attempted to summarize the anti-inflammatory and antioxidative mechanisms of action exerted by SGLT2i and GLP1RA in the kidneys and other tissues and organs in the setting of diabetes and other models of injury.

## 7. Summary and Conclusions

The scientist should always remain skeptical and doubtful. ‘Dubito ergo sum’ is the principle of approaching medical data and reading medical papers. While reading medical literature on SGLT2i and GLP1RA, we asked ourselves: ‘could this happen?’, ‘is it possible to obtain such great results?’ and ‘is it okay to become more and more enthusiastic about these data?’. We were concerned, but finally we came to the conclusion that the holy grail in the treatment of T2D and its complications, and very likely in the treatment of many other diseases, has just been found. Too much data clearly and consistently point to the effectiveness of SGLT2i and GLP1RA in the treatment of the discussed conditions to remain doubtful. The results of large clinical trials performed on humans perfectly correspond with evidence from basic science studies highlighting the mechanisms of clinical benefits for patients: for their hearts, kidneys, vessels, livers, brains and for their lives.

Since new classes of glucose-lowering drugs have emerged and the results of large outcome trials with their use have been announced, the standard treatment choices for T2D have changed. The non-glycemic properties of both drug classes were so strongly pronounced that for the first time in the history of diabetology, the treatment adjustment need not necessarily be based on ‘metabolic’ parameters. Until the year 2019, the main reason for adding another antihyperglycemic drug or changing to another drug class was the HbA1c value exceeding the guideline-defined value; now, the treatment decisions take into account the co-existing CKD and cardiovascular disease [134].

SGLT2i and GLP1R agonists open new avenues to the successful treatment of patients with diabetes, focusing not only on glycemic targets, but also cardiovascular, cerebrovascular and renal outcomes (which will translate into improved survival and preserved quality of life). New drugs will hopefully allow for the opportunity of such a beneficial outcome to most patients with T2D, but it must be admitted that using more ‘traditional’ but structured and multifactorial interventions with close monitoring to detect adverse effects of therapy may also allow spectacular results to be achieved. For example, the Italian multicenter ‘Nephropathy in the Diabetes Type 2′ (NID-2) study group randomly applied the strict and closely monitored strategy of achieving a blood pressure goal of <130/80 mmHg, HbA1c < 7%, total cholesterol <175 mg/dL and HDL-cholesterol >40 mg/dL for men and >50 mg/dL for women vs. the standard of care in albuminuric T2D diabetic patients without previous MACE. All mentioned goals of treatment were achieved statistically more frequently in patients treated according to the multifactorial intervention protocol, and this difference also translated into an astonishing reduction in both MACE and all-cause death by 50% over a 13-year observation period [135]. Such a multifactorial, multitarget, patient-centered approach should never be abandoned in the treatment of T2D even in the era of ‘newer’ drugs, although it must be admitted that one of the strategies used in the NID-2 trial (namely, a combination of ACEi and ARB) is no longer recommended in most of the current guidelines.

## Figures and Tables

**Figure 1 ijms-22-10822-f001:**
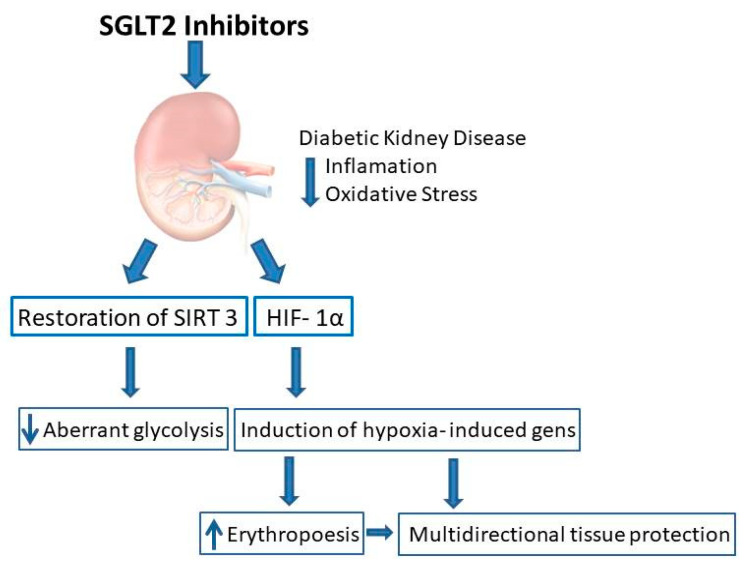
Interaction between SGLT2 inhibition, HIF1α and tissue protection in diabetic kidney disease.

**Figure 2 ijms-22-10822-f002:**
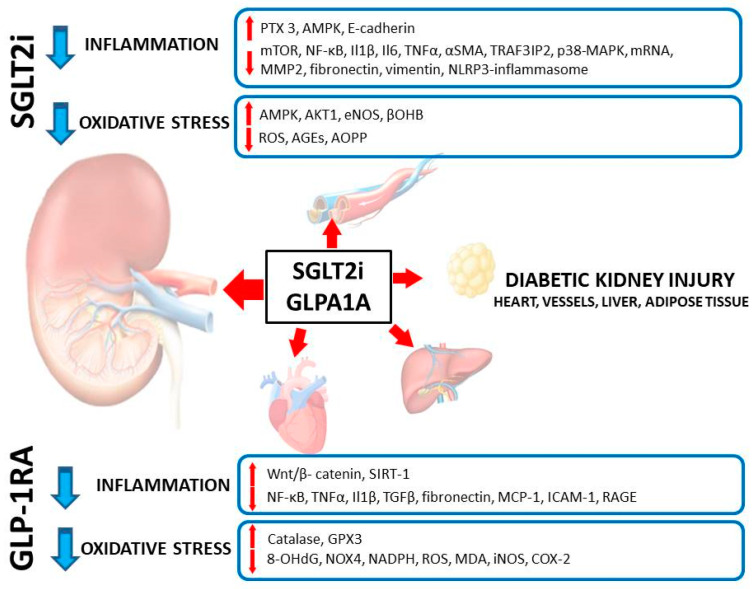
Selected mechanisms of interaction between SGLT2i, GLP1RA and inflammation/oxidative stress in the kidney and other target organs/tissues. Legend to Figure 2: 8-OHdG—8-Hydroxy-deoxyguanosine; AGEs—Advanced glycation end products; AKT1—AKT Serine/threonine kinase 1; AMPK—5′AMP-activated protein kinase; AOPP—Advanced oxidation protein products; COX-2—Cyclooxygenase-2; eNOS—*Endothelial nitric oxide synthase*; GPX3—Glutathione Peroxidase 3; ICAM-1—Intercellular adhesion molecule 1; Il1β—Interleukin1β; Il6—Interleukin 6; iNOS—*Inducible nitric oxide synthase*; MCP-1—*Monocyte chemoattractant protein-1*; MDA— malondialdehyde; MMP2—Matrix metalloproteinase-2; mRNA—*messenger RNA*; mTOR—Mammalian target of rapamycin; NADPH—Nicotinamide adenine dinucleotide phosphate; NF-κB—nuclear factor kappa-light-chain-enhancer of activated B cells; NLRP3-inflamassome—NOD-, LRR- and pyrin domain-containing protein 3 inflammasome; NOX4—NADPH Oxidase 4; p38-MAPK—*p38* Mitogen-activated protein kinase; PTX 3—Pentraxin 3; RAGE—Receptor for advanced glycation end products; ROS—Reactive oxygen species; SIRT—Sirtuins; TGFβ—Transforming growth factor β; TNFα—tumor necrosis factor α; TRAF3IP2—Tumor necrosis factor *receptor associated factor 3* interacting protein 2; αSMA- α-Smooth muscle actin; βOHB—β-*Hydroxybutyric acid*.

**Table 1 ijms-22-10822-t001:** Selected mechanisms of interactions between SGLT2 inhibitors, inflammation and oxidative stress.

	Inflammation	Oxidative Stress
	**Kidney**	
**SGLT2i**	- Promote autophagic potential of renal cells (induction of AMPK phosphorylation AMPK) [24] - Decrease activity of mammalian target of rapamycin (mTOR) [24,68] - Decrease synthesis of proinflammatory cytokines: interleukin 1β (IL1β), interleukin 6 (IL6) and tumor necrosis factor α (TNFα) through inhibition of NF-κB [24,25,35,37,66,73,82] - Suppress p65 subunit of NF-κB, secondary to activation of AMPK [24,25,26,35] - Prevent expression of αSMA (alpha-smooth muscle actin), a marker of cell de-differentiation that inhibits changes of the phenotype of tubular cells into the proinflammatory and profibrotic [25] - Partially restore expression of pentraxin 3 (PTX 3) (PTX3 favorably attenuates inflammatory activity of macrophages, promotes M2 phenotype and downregulates NF-κB, IL1β, TNFα and monocyte chemoattractant protein 1 (MCP1)) [24,26,65] - Inhibit TRAF3IP2 (TRAF3-interacting protein 2), the proinflammatory protein induced by high glucose, activating IκB kinase (IKK)/NF-κB and promoting the expression of inflammatory mediators; prevent advanced glycation end-product-mediated stimulation of TRAF3IP2 expression [35] - Reduce activity of p38-MAPK [35] - Prevent high-glucose-induced synthesis and release of matrix metalloproteinase 2 (MMP2) [35] - Prevent upregulation of mesenchymal phenotype, i.e., αSMA, fibronectin and vimentin and downregulation of E- cadherin [35] - Reduce renal expression of mRNA of collagens type 1 and type 3 [37] - Decrease the activity of NLPR3 inflammasome and caspase-3 and increase the phosphorylated/total AMPK ratio [37,38,72] - Decrease caspase-1 activation [38]	- Prevent increase in reactive oxygen species [22] - Abolish the superoxide generation in tubular cells [35] - Augment the antioxidant defense mechanisms, activate AMPK, AKT serine/threonine kinase 1 (AKT1) and eNOS [39,72] - Prevent formation of advanced glycation and oxidation products [39] - Increase in β-hydroxybutyric acid (βOHB) [41]
	**Other organs/tissues/cells**	
**SGLT2i**	- Liver: decrease the amount of MDA, interleukins 1β and 18 and TNFα [54,55,58] - Liver: promote autophagy, inhibit apoptosis and inflammation (by reducing mTOR signaling and MCP-1 expression) [57] - Liver and adipose tissue: decrease expression of mRNA for such mediators of p38-mitoge-activated protein kinase [p38-MAPK], NF-κB or extracellular signal-regulated kinase [ERK]) [66] - Adipose tissue: increase autophagic flux by upregulation of sirtuin 1 (SIRT1), fibroblast growth factor 21 (FGF-21) and peroxisome proliferator-activated receptor γ co-activator 1α (PGC1α); upregulation of SIRT1 leads to activation of hypoxia-inducible factor 2α (HIF2α) [60,61,62,63,64] - Adipose tissue: inhibit phosphorylation of NF-κB and signal transducer and activator of transcription (STAT) 1 and 3, Janus (JAK2) kinase and IKK [66] - Adipose tissue: downregulate IKK/NF-κb, MKK7/JNK and JAK2/STAT1 pathways [66] - Endothelial cells: inhibit hexokinase 2 (HK2) activation, HK2-mediated ERK1/2 phosphorylation and IL-6 synthesis and increase AMPK activation in LPS [76]	- Liver: reduced oxidative stress by means of decreased level of MDA and sustained enzymatic activity of superoxide dismutase [54,58] - Liver: prevent downregulation of Nrf2 and PPARγ (peroxisome proliferator-activated receptor gamma) mRNA [55,56] - Cardiomyocytes: augment nuclear factor erythroid 2-related factor 2/heme oxygenase 1 (Nrf2- and HO-1)-mediated antioxidant and anti-inflammatory signaling [40] - Cardiomyocytes: upregulate Nrf2/HO-1 (nuclear factor-erythroid 2 related factor/heme oxygenase 1) [68] - Cardiomyocytes: reduce concentration of H_2_O_2_, 3-nitrotyrosine and lipid peroxide in cytosol and mitochondria [43] - Cardiomyocytes: reduce oxidation of PKGIα protein kinase G type Iα, the cGMP-dependent protein kinase controlling the calcium flux within cells [43] - Adipose tissue: reduce 8-iso-prostaglandin F2α and 8-hydroxy-2′-deoxyguanosine [8-ohdg] [82]

**Table 2 ijms-22-10822-t002:** Selected mechanisms of interactions between GLP-1R agonists, inflammation and oxidative stress.

	Inflammation	Oxidative Stress
	**Kidney**	
**GLP-1R agonists**	- Suppress NF-κB [89] - Normalize expression of TNFα, IL-1β, TGFβ, fibronectin, inhibit chemokines attracting macrophages to the site of injury and promoting their infiltration (MCP-1, ICAM-1) [88,89,91,92,93,96,102,104] - Inhibit transition of cultured mesangial cells into fibroblast-like phenotype (with concomitant inhibition of TGFβ1-signaling) [92] - Downregulate RAGE expression on mesangial cells [96] - Increase the Wnt/β-catenin signaling resulting in reduction of ECM protein synthesis [98] - Reverse downregulation of SIRT-1 [99] - Decrease the mRNA expression of collagen type I alpha 1 chain (the marker of fibrosis), reduce the renal infiltrates with PMN, macrophages, T-helper cells and cytotoxic T lymphocytes; reduce gene expression of IFγ, TNFα and T-bet [104,105]	- Decrease 8-hydroxydeoxyguanosine content, mRNA of NADPH 4 oxidase (NOX4) and NOX4 protein expression [88,89,93,100] - Inhibit NADPH oxidase activity [93] - Decrease reactive oxygen species generation and MDA production [89,91,93] - Restore content of catalase and glutathione peroxidase-3 [90] - Decrease expression of renal iNOS and COX-2 [91]
	**Other organs/tissues/cells**	
**GLP-1R agonists**	- Adipocytes: decrease levels of IL-6 and MCP1 mRNA [108] - Adipocytes: increase translocation of the transcription factor Forkhead Box O1 (Foxo1) from cytoplasm to the nucleus [109] - PBMC: suppress secretion of chemokines CCL5 and CXCL10 [110] - PBMC: decrease levels of phosphorylated kinases: phos-ERK and phos-p38 MAPK [110] - PBMC: suppress NFκB, reduce mRNA of TNFα and IL-1β [111,112,116] - PBMC: suppress mRNA of c-JUN N- terminal kinase (JNK1), toll-like receptors (TLR) 2 and 4 [111] - Monocytes: reduce expression of mRNA for ICAM1, VCAM1, TNFα and MCP1 in these cells [112] - Macrophages: promoted an activation of signal transducer and activator of transcription 3 (STAT3), followed by increased expression of IL-10 [114,116] - Macrophages: reverse macrophage shift from M1 towards M2 [116] - Decrease number of peripheral regulatory T cells (Treg) [115] - Cardiomyocytes/liver: promote autophagy—activation (phosphorylation) of AMPK and inactivation of mTOR [117,118] - Cardiomyoblasts: increase cAMP content in the cells [119]	- PBMC: decrease ROS generation [111] - Cardiomyoblasts: decrease mitochondrial ROS generation and stabilizes mitochondrial membrane potential [119] - Cardiomyoblasts: decrease reactive oxygen species production [123] - Cardiomyocytes: increase expression of MnSOD and catalase [123]

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
