# Peer review of "Inflammation and Oxidative Stress in Diabetic Kidney Disease: The Targets for SGLT2 Inhibitors and GLP-1 Receptor Agonists"

_ijms, 2021, doi:10.3390/ijms221910822_

Round 1

Reviewer 1 Report

The manuscript is interesting. The topic is hot, even if it has been discussed in previous reviews. The references are quite up to date.

This reviewer raises some comments to enrich the text and conclusions

1- Very recently has been showed a role for SGLT2/SIRT6 pathway in the inflammatory process of diabetic atherosclerotic lesions. These findings suggest a possible favorable modulation of this mechanism by SGLT2i (Mol Metab. 2021 Sep 6:101337. doi: 10.1016/j.molmet.2021.101337.). This issue as well as this reference should be added in the manuscript.

2- Very recent reviews and meta-analyzes have confirmed the protective role of SGLT2i against heart failure (International Journal of Molecular Sciences, 2021, 22(11), 5863. doi: 10.3390/ijms22115863) and stroke risk-related atrial fibrillation (Stroke, 2021, 52(5), pp. 1545–1556. doi: 10.1161/STROKEAHA.120.031623). These very interesting problems for the reader should be commented on and above references added in the introduction of the authors.

3- In Conclusions the authors state “Until the year 2019 the main reason for 1188 adding another antihyperglycemic drug or changing to another drug class was the HbA1c value exceeding guideline-defined value; now the treatment decisions take into account the co-existing CKD and cardiovascular disease”. Of course, I agree with this conclusion, which simply reiterates the claims of the recent ADA and EASD guidelines. Since primary and secondary prevention of morbidity and mortality (especially cardiovascular) is at the center of modern therapy for diabetic nephropathy and diabetes in general, I believe it is appropriate to underline that very recently this goal has been reached for the first time by randomized multicenter NID study through a multifactorial intervention in albuminuric type 2 diabetic subjects, without previous MACE (Cardiovasc Diabetol (2021) 20:145. doi: 10.1186/s12933-021-01343-1). This strategy, in addition to the use of multitarget drugs such as SGLT2i and GLP-1 RA, represents the future of the therapy of diabetic patients. From my point of view, this essential issue must be commented on in conclusions.

4- The two tables are clear, but a figure that schematizes the mechanisms of action of SGLT2i and GLP-1 RA described by the authors would help readers.

5- The manuscript should be reviewed by a native English speaker.

Author Response

Answers to Reviewer 1:

Thank you very much for all of your important comments. I am sure you will agree with us that nowadays SGLT2i and GLP1R agonists have become a moving target – the apparently updated review may become outdated by the time of publication. Nevertheless, we appreciate you focusing our attention on these important papers.

  1. We discussed in detail the landmark study of D’Onofrio and colleagues published in Mol Metabol. 2021.
  2. We added the suggested citation from In J Mol Sci and Stroke journal and some comments on cardiovascular and cerebrovascular protection exerted by SGT2i in issues discussed by these papers. We also decided to add the newest publications on heart failure as well as the metaanalysis of such trials. We also mentioned the recent ESC guidelines on heart failure, incorporating SGLT2i data.
  3. We read with great interest the paper of Sasso et al. from Cardiovasc Diabetol. 2021. We summarized key results of this trial and emphasized the need for a multifactorial approach to patients with diabetes – we share your opinion that such an approach should always be applied.
  4. We did our best to address your suggestion concerning the figure.
  5. Now the paper has been read by the native American English speaker with a medical and basic science background and we do believe its quality has been improved.

Thank you again for your kind comments and suggestions

Yours sincerely,

Tomasz Stompór, M.D.

Professor of Medicine

Reviewer 2 Report

The authors describe the comprehensive analysis of oxidative stress and inflammation in DKD and discuss the effect of SGLTi and GLP1RA in mitigating inflammation and oxidative stress. The information in the manuscript is outdated. I suggest the authors including new researches and improve the manuscript based on the following suggestions.

  • Differences in the protective abilities of different classes of SGLT-2i and GLP-1 R agonists in renal outcome in diabetes. Which class of medication is better treating against DKD?
  • The authors did not describe unfavorable effects of SGLTi and GLP1RA observed during small scale and large scale RCTs
  • Introduction: Other available medications against DKD such as statins, endothelin antagonists, mineralocorticoid receptor antagonists and blood pressure lowering agents. Potential drugs have been identified such as DPP-4 inhibitor linagliptin, SIRT3, JAK-stat inhibitors, glycolysis inhibitors, and ACE inhibitors, ARBs, and peptide AcSDKP in protecting DKD.
  • Endothelial glucocorticoid receptors and podocyte glucocorticoid receptor in combating diabetic nephropathy.
  • Do SGLT-2i have SGLT-2 independent effects? And similarly describe about GLP-1 RA?
  • Describe abnormal glycolysis in DKD. HIF1a level and DKD.
  • SGLT2i elevates SIRT3 level, which protects the kidneys in two ways- by suppressing the abnormal glycolysis and restoring fatty acid oxidation. Hence SGLT2i improves cellular and mitochondrial health as well. I would suggest the authors to include a separate section with figure.
  • Mechanisms of oxidative stress and inflammation; role of elevated levels of proinflammatory cytokines in EMT and EndMT. Effect of SGLT-2i and GLP-1RA in suppressing the levels of pro-inflammatory cytokines and inhibiting EMT and EndMT processes. Include a figure.   
  • SGLT2i and autophagy defects
  • Effects of SGLT-2i and GLP1RA in hemodynamic alterations and RAAS.
  • GLP1RA and DPP-4
  • Pathogenic role of DPP-4 in DKD; DPP-4 inhibitor Linagliptin as a potential drug combating DKD. Discuss it. Linagliptin elevates endogenous antifibrotic peptide AcSDKP in diabetes. AcSDKP is essential peptide important for kidney cell health by inhibiting TFGB-signaling, improve cytokine health and restores the antifibrotic microRNAs crosstalk mechanism in diabetic endothelial cells. Discuss it.
  • microRNAs and long non-coding RNAs in EMT and EndMT. Impact of SGLT-2i and GLP1RA?

Author Response

Answers to the Reviewer 2:

Thank you very much for all of your important comments. We appreciate the high scores we received from you in the respective categories. Let us address point-by-point your comments:

  1. Let us mention that we decided to write the review discussing basic science data. In both the Introduction and Summary sections, we mention the key studies and clinical aspects of using SGLT2i and GLP1R agonists in the treatment of diabetes and DKD. Please note that in its original form the manuscript had 42 pages and following R1 it has already been expanded to 52 pages. Trying to describe both groups of drugs in two aspects (inflammation and oxidative stress) in relation to the kidney and out-of-the kidney tissues was a big challenge. Adding clinical data, data on other drugs you mention (such as DPP4i, JAK-stat inhibitors, glycolysis inhibitors, glucocorticosteroid receptors on endothelial cells and podocytes ) would be clearly beyond the scope of our paper and would result in writing a book rather than a review paper. Adding this information would make our manuscript indigestible for potential readers. Please note that the role of ACEi and ARBs in DKD is a ‘canonical’ knowledge and we do not think that discussing it would add any value to our paper. For the same reason we think that describing the relation between DPP4i and GLP1RA, the role of DPP in the development of DKD, interactions between SGLR2i, GLP1-R agonists, microcirculation and RAAS system, or a detailed discussion on linagliptin would be beyond the scope of our paper, especially giving the fact that DPPi failed to prove their efficacy in CVOT. For the same reason (out of the scope), a detailed description of clinical benefits and side effects of discussed drugs was not undertaken.
  2. We discussed the impact of respective drugs on autophagy and the SIRT system. We did our best to identify most of the important studies in the field but since the knowledge concerning SGLT2i and GLP1RA is developing so rapidly, we obviously were unable to identify all of the facts, and for the sake of the paper size we needed to choose only some of them. Nevertheless, we added a discussion on one of the landmark papers in the field that describes the relationship between SIRT and SGLT2i and we mentioned such an interaction in a couple of other places throughout our review.
  3. We appreciate attracting our attention to the impact of SGLT2i and GLP1-R agonists on EMT and EndMT. Although we mentioned such an impact in our original submission, now we have expanded this topic.
  4. The same appreciation applies to the interaction between SGLT2i and GLP1-R agonists and miRNA. We also added some comments and cited important studies.
  5. As you suggested we added comments and citations on aberrant glycolysis and the interaction of SGLT2i with the HIF pathway; we also added a corresponding figure.
  6. Now the paper has been proofread and corrected by the native American English speaker with a medical and basic science background and we do believe its quality has been improved.

Thank you again for your kind comments and suggestions. We hope you understand our approach to the paper and the initial idea of its editing and structuring. We hope now you will find it acceptable.  

Yours sincerely,

Tomasz Stompór, M.D.

Professor of Medicine

Round 2

Reviewer 1 Report

All issues raised by this reviewer were addressed by the authors. No further comment.

Reviewer 2 Report

Ref 32 and 78 are same. 

Correct it.